# HopE and HopD Porin-Mediated Drug Influx Contributes to Intrinsic Antimicrobial Susceptibility and Inhibits Streptomycin Resistance Acquisition by Natural Transformation in *Helicobacter pylori*

Yixin Liu,[a,b,c] (ID) Feng Yang,[a,b,c] Su Wang,[a] Wenjing Chi,[a] Li Ding,[a] Tao Liu,[a] Feng Zhu,[a] Danian Ji,[d] Jun Zhou,[d] Yi Fang,[a] Jinghao Zhang,[a] Ping Xiang,[d] Yanmei Zhang,[a,b,c] Hu Zhao[a,b,c]

[a]Department of Laboratory Medicine, Huadong Hospital, Fudan University, Shanghai, China
[b]Shanghai Key Laboratory of Clinical Geriatric Medicine, Shanghai, China
[c]Research Center on Aging and Medicine, Fudan University, Shanghai, China
[d]Department of Endoscopy, Huadong Hospital, Fudan University, Shanghai, China

**ABSTRACT** *Helicobacter pylori* is a human pathogen competent for natural transformation. Intrinsic and acquired antibiotic resistance contribute to the survival and multiplication of *H. pylori* under antibiotics. While drug-resistance dissemination by natural transformation (NT)-mediated horizontal gene transfer remains poorly understood in *H. pylori*. The purpose of the study was to investigate the role of *H. pylori* porins (HopA, HopB, HopC, HopD, and HopE) in the intrinsic antibiotic resistance and to preliminarily reveal the potential effect of HopE and HopD porins in streptomycin resistance acquisition after NT in the presence of antibiotics. Using traditional antibiotic susceptibility tests and growth curve analysis, we found the MIC values of metronidazole, clarithromycin, levofloxacin, tetracycline, rifampin, and streptomycin in mutants lacking HopE and/or HopD were significantly elevated compare to those in wild-type strain. The quantitative analysis of the tetramethyl rhodamine isothiocyanate (TRITC)-labeled streptomycin accumulation at the single-cell level showed reduced streptomycin intracellular fluorescence in Δ*hopE* and Δ*hopD* mutant cells. Furthermore, in the presence of translation-inhibiting antibiotic streptomycin, the resistance acquisition frequency was decreased in the wild-type strain, which could be reversed by mutants lacking HopE and HopD that restored relatively high resistance acquisition frequencies. By transforming a pUC19-*rpsL*mut-*sfgfp* linear plasmid carrying a streptomycin conferring mutation, we observed that the impaired ability of rpsLmut synthesis in the wild-type strain was restored in the Δ*hopE* and Δ*hopD* mutant transformants. Our study revealed that in the presence of streptomycin, resistance acquisition at least partially relied on the deletion of the *hopE* and *hopD* genes, because their loss reduced streptomycin concentration in the cell and thus restored the expression of the resistance-conferring gene, which was inhibited by streptomycin in wild-type strain. The loss of HopE and HopD influx activity may also preserve resistance acquisition by transformation in the presence of antibiotics with other modes of action.

**IMPORTANCE** *Helicobacter pylori* is constitutively competent for natural transformation (NT) and possesses an efficient system for homologous recombination, which could be utilized to study the NT-mediated horizontal gene transfer induced antibiotic resistance acquisition. Bacterial porins have drawn renewed attention because of their crucial role in antibiotic susceptibility. From the perspective of porin-mediated influx in *H. pylori*, our study preliminarily revealed the important role of HopE and HopD porins not only in preserving the intrinsic susceptibility to specific antibiotic but also in evading acquired antibiotic resistance by NT in the presence of translation-inhibiting antimicrobial. Therefore, the loss of HopE or HopD porin in *H. pylori* genomes,

Address correspondence to Yanmei Zhang, 15618653286@163.com, or Hu Zhao, ZH13701618011@163.com.

The authors declare no conflict of interest.

combined with the large number of secreted or cell-free genetic elements carrying mutations conferring antibiotic resistance, may raise the possibility that this mechanism plays a potential role in the propagation of antibiotic resistance within *H. pylori* communities.

**KEYWORDS** *H. pylori*, HopE, HopD, porin, antibiotic resistance, natural transformation, *Helicobacter pylori*

The survival and multiplication of bacteria under selective pressures of antibiotics rely on intrinsic and acquired antibiotic resistance (1, 2). In recent years, the emergence and dissemination of antibiotic resistance within pathogens has produced public apprehension, which has drawn much attention in microbiology research. Horizontal gene transfer (HGT)-induced antibiotic resistance gene (ARG) acquisition of bacteria is the main reason for the propagation of drug resistance, which includes three main mechanisms: conjugation, transduction, and natural transformation (3). Unlike conjugation and transduction, natural transformation (NT) is a pathway by which bacteria import DNA from the extracellular environment and internalize it into the cytoplasm, followed by either integration into the bacterial genome or recircularization (in the case of plasmid DNA) (3, 4). Although NT is relevantly less studied at present, it has potential that cannot be underestimated due to the ability of NT to transfer ARGs encoded by both plasmids and genomic DNA (3, 5). Moreover, in 2017, the World Health Organization (WHO) published a series of antibiotic-resistant pathogens in urgent need of effective antibiotics (6). We noticed that more than half of these pathogens are engaged in NT-mediated DNA intake, which underlined the importance of investigating multiple factors affecting HGT-induced propagation of antibiotic resistance.

The human pathogen *Helicobacter pylori* impacts more than half of the world's population and is closely associated with multiple digestive system diseases such as gastritis, peptic ulcer, gastric cancer, and gastric mucosa-associated lymphoid tissue (MALT) lymphoma (7, 8). However, *H. pylori* manifests increasing antibiotic resistance (AR) rates, which has weakened the eradication efficacy (9). Multiple lines have demonstrated that *H. pylori* is constitutively competent for NT and possesses a set of machinery proteins for efficient HGT and homologous recombination (10), which enables dissemination of antimicrobial resistance by NT in *H. pylori*. For example, there have been some reports showing coinfection with multiple *H. pylori* strains with different antibiotic resistance patterns in a single patient (11) and regions with both high *H. pylori* prevalence and high chance of multiple infection with different strains, especially developing countries (more than 20 to 35%) (12–15). Furthermore, resistance to metronidazole has been positively correlated with capacity for NT in *H. pylori* clinical isolates (16). In addition, NT enables *H. pylori* strains to transmit streptomycin resistance-conferring DNA fragment (10, 17, 18).

Bacterial porins have drawn renewed attention because of their crucial role in antibiotic susceptibility (19). Currently, in *H. pylori*, at least five $\beta$-barrel outer membrane proteins (HopA, HopB, HopC, HopD, and HopE) of Hop members have been characterized as porins, which have structural homology with the *Escherichia coli* outer membrane protein F (OmpF) porin (20), with the main candidate nonspecific HopE porin forming large channels (21). Importantly, porins are the preferred route for the entry of some antibiotics such as $\beta$-lactams and fluoroquinolones in Gram-negative pathogens (22). The clinical relevance of porin defects and/or multidrug efflux pump overexpression forming membrane-associated resistance mechanisms has been well established for these antibiotics (23). Many studies reported that a large number of pathogens developed antibiotic resistance through loss or modification of porins, and these bacteria were but not limited to *E. coli*, *Neisseria gonorrhoeae*, *Pseudomonas aeruginosa*, *Klebsiella pneumoniae*, and *Enterobacter aerogenes* (24). Moreover, a previous study also reported that *H. pylori* HopE porin presented scarce gene presence and expression among clinical isolates (20). However, the role of porins in intrinsic and acquired resistance in *H. pylori* has yet to be described.

In this study, we preliminarily demonstrated that HopE and HopD porins performed influx of streptomycin, the defect of which enhanced the intrinsic antibiotic tolerance represented by increased MIC values and, in the presence of translation-inhibiting antibiotic streptomycin, had a restoring effect on the resistance phenotype conversion and expression of newly acquired resistance-conferring gene, which were inhibited in the wild-type strain after the plasmid naturally transformed.

## RESULTS

**The MICs of the porin-deleted *H. pylori* mutants.** The reference strain *H. pylori* 26695 (wild type) and five porin-deleted mutants (Δ*hopA*, Δ*hopB*, Δ*hopC*, Δ*hopD*, and Δ*hopE*) were tested for MICs against seven antibiotics: metronidazole, clarithromycin, levofloxacin, amoxicillin, tetracycline, rifampin, and streptomycin, using ETEST and/or agar dilution methods. There was no colony growth observed on the Mueller-Hinton (MH) plates tested for tolerance of wild-type (WT), Δ*hopA*, Δ*hopB*, and Δ*hopC* strains to all seven antibiotics (Fig. 1A). In the ETEST assessing the antibiotic tolerance of Δ*hopD* and Δ*hopE* mutants, all the plates manifested visible inhibition zones (Fig. 1A, Δ*hopD* and Δ*hopE* lanes). On the plates assessing Δ*hopE* tolerance to metronidazole, levofloxacin, and rifampin, the MIC ranges were 2 to 4, 0.08 to 0.16, and 0.125 to 0.25 $\mu$g/mL, respectively. On the plates assessing Δ*hopD* resistance to levofloxacin and rifampin, the MICs were 0.08 to 0.16 and 0.25 to 0.5 $\mu$g/mL, respectively. However, the remaining seven plates did not show inhibition zones intersecting with the corresponding strips. To obtain the specific MICs of each antibiotic on the porin-deleted mutants, the agar dilution method (0.001 to 32 $\mu$g/mL) was employed. The MICs of levofloxacin, tetracycline, and streptomycin on both Δ*hopD* (0.052 $\pm$ 0.018 $\mu$g/mL, $P = 0.0097$; 0.042 $\pm$ 0.018 $\mu$g/mL, $P = 0.0303$; and 5.333 $\pm$ 2.309 $\mu$g/mL, $P = 0.0168$) and Δ*hopE* (0.084 $\pm$ 0.036 $\mu$g/mL, $P = 0.0173$; 0.084 $\pm$ 0.016 $\mu$g/mL, $P = 0.0201$; and 6.667 $\pm$ 2.309 $\mu$g/mL, $P = 0.0078$) mutants were significantly higher than those on the WT strain (0.003 $\pm$ 0.001, 0.006 $\pm$ 0.003, and 0.062 $\pm$ 0.054 $\mu$g/mL), respectively (Fig. 1C, G, and H). Moreover, the MICs of metronidazole and clarithromycin on the Δ*hopE* mutant (3.333 $\pm$ 1.155 $\mu$g/mL, $P = 0.0075$; and 0.042 $\pm$ 0.018 $\mu$g/mL, $P = 0.0312$) were significantly higher than those on the WT strain (0.002 $\pm$ 0, 0.007 $\pm$ 0.002, and 0.003 $\pm$ 0.001 $\mu$g/mL), respectively (Fig. 1B and D). The MIC of rifampin on the Δ*hopD* mutant (0.583 $\pm$ 0.382 $\mu$g/mL, $P = 0.0052$) was significantly higher than that on the WT strain (0.003 $\pm$ 0.001 $\mu$g/mL) (Fig. 1F).

**HopD and HopE depletions significantly enhance the maximal growth of *H. pylori* in the presence of antibiotics.** We further determined the growth of Δ*hopD* and Δ*hopE* mutants in the presence of the indicated concentrations (Table S1) of antibiotics at 80 h of incubation. In the presence of metronidazole (1 $\mu$g/mL), levofloxacin (0.01 and 0.1 $\mu$g/mL), rifampin (0.1 $\mu$g/mL), and streptomycin (1 and 5 $\mu$g/mL), the growth biomasses of both Δ*hopD* ($P = 0.0013$, $P < 0.0001$, $P = 0.0005$, $P = 0.0003$, $P = 0.0023$, and $P < 0.0001$, respectively) and Δ*hopE* ($P < 0.0001$, $P = 0.0002$, $P = 0.0009$, $P = 0.0002$, $P = 0.0011$, and $P < 0.0001$, respectively) mutants were significantly higher than that of the WT strain (Fig. 1J, L, M, R, T, and U). In the presence of metronidazole (4 $\mu$g/mL), clarithromycin (0.05 $\mu$g/mL) and tetracycline (0.1 $\mu$g/mL), the growth biomasses of Δ*hopE* mutants ($P = 0.0001$, $P = 0.0082$, and $P < 0.0001$, respectively) were significantly higher than that of the WT strain (Fig. 1K, N, and V). These results implicated that HopD and HopE deletions contributed to a degree of intrinsic tolerance to subinhibitory concentrations of a range of antibiotics. Notably, among the six antibiotics tested, four (rifampin, clarithromycin, tetracycline, and streptomycin) are protein synthesis-inhibiting antibiotics.

**HopE and HopD perform influx of streptomycin to affect the uptake of the antibiotic in *H. pylori*.** Since HopD and HopE are porins forming large channels across cell membranes (21), we hypothesized that the decreased antibacterial effect of antibiotics may be due to the reduced intake of the antibiotic. To test this, we detected the antibiotic uptake by WT, Δ*hopA*, Δ*hopB*, Δ*hopC*, Δ*hopD*, and Δ*hopE* strains when incubation was performed in the presence of tetramethyl rhodamine isothiocyanate (TRITC)-labeled streptomycin. The intracellular fluorescence of TRITC-labeled streptomycin in Δ*hopA*,

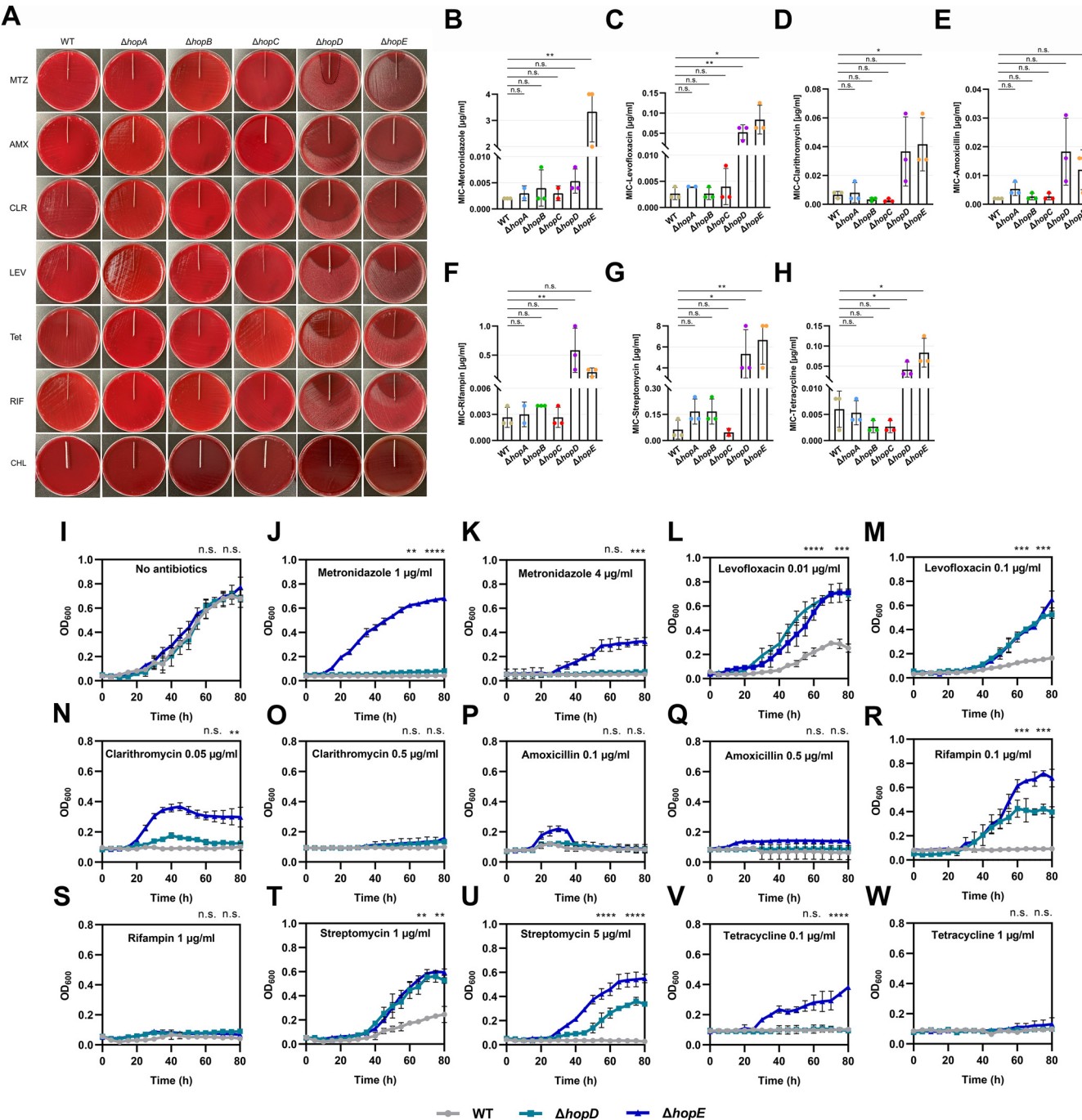

**FIG 1** MICs of antibiotics in five porin-deletion *H. pylori* mutants and growth of Δ*hopD* and Δ*hopE* mutants in the presence of indicated concentrations of antibiotics. (A) The wild-type (WT) strain *H. pylori* 26695 and Δ*hopA*, Δ*hopB*, Δ*hopC*, Δ*hopD*, and Δ*hopE* mutants were examined for MICs of metronidazole (MTZ), clarithromycin (CLR), levofloxacin (LEV), amoxicillin (AMX), tetracycline (Tet), rifampin (RIF), and chloramphenicol (CHL) by ETEST. The inhibition zones were observed at 72 h. The MIC values were defined by the point of intersection of the inhibition ellipse zone with the graded strips for the ETEST. (B to H) WT and five porin-deleted strains were examined for MICs of metronidazole, clarithromycin, levofloxacin, amoxicillin, tetracycline, rifampin and streptomycin by the agar dilution method. The values are shown as means ± standard deviation (SD) for two or three biologically independent experiments. *, significant ($P < 0.05$); **, very significant ($P < 0.01$); n.s., not significant. (I to W) Growth curves of WT, Δ*hopD*, and Δ*hopE* strains growing in *Brucella* broth in the absence or presence of metronidazole (1 and 4 $\mu$g/mL), amoxicillin (0.1 and 0.5 $\mu$g/mL), clarithromycin (0.05 and 0.5 $\mu$g/mL), levofloxacin (0.01 and 0.1 $\mu$g/mL), tetracycline (0.1 and 1 $\mu$g/mL), rifampin (0.1 and 1 $\mu$g/mL), and streptomycin (1 and 5 $\mu$g/mL). The values are shown as means ± SD for independent triplicates. The significance analysis results above the images were from Δ*hopD* versus WT and Δ*hopE* versus WT at 80 h, respectively. ****, highly significant ($P < 0.0001$); ***, highly significant ($P < 0.001$); **, very significant ($P < 0.01$); *, significant ($P < 0.05$); n.s., not significant. The source data for panels A to W are provided as a source data file. $OD_{600}$, optical density at 600 nm.

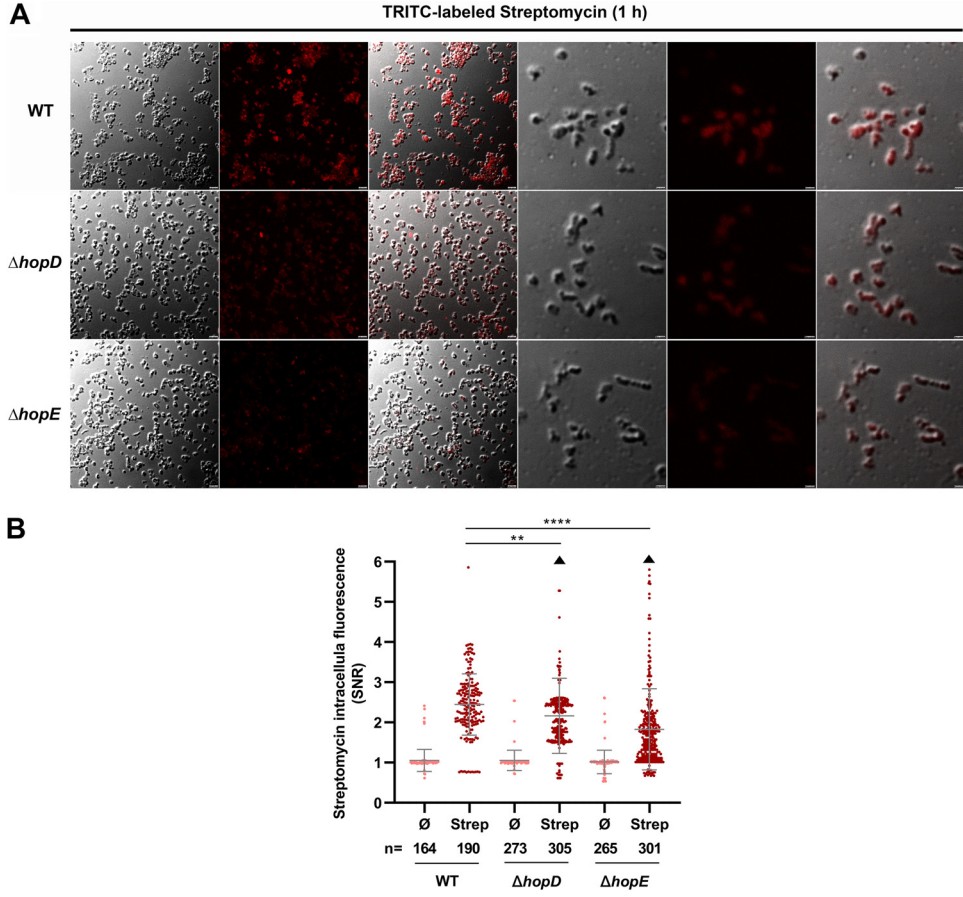

**FIG 2** HopD and HopE perform the uptake of streptomycin. (A) Representatives of red fluorescence detection after 1 h of incubation in the presence of tetramethyl rhodamine isothiocyanate (TRITC)-labeled streptomycin. An automatic red fluorescence channel and merged images of TRITC with differential interference contrast (DIC) are shown. The scale bars correspond to 2.5 and 1 $\mu$m, respectively. (B) Quantification of streptomycin (Strep) uptake by WT, $\Delta hopD$, and $\Delta hopE$ strains. TRITC-labeled streptomycin allows for quantification of the amount of drug that penetrates the cells. Scatter-dot plots show the single-cell quantification of intracellular fluorescence (signal/noise ratio, SNR) in the population of cells after 1 h of incubation with streptomycin (5 $\mu$g/mL). The median, quartile 1, and quartile 3 are indicated by horizontal lines. Black dots above and below the maximum and minimum values correspond to outlier cells. Triangles indicate the presence of data points beyond the axis range. The number of cells analyzed (n) is also indicated. ****, highly significant ($P < 0.0001$); **, very significant ($P < 0.01$); *, significant ($P < 0.05$). The source data are provided as a source data file.

$\Delta hopB$, and $\Delta hopC$ mutants all showed unchanged levels compared with that in the WT strain (Fig. S1), while we observed that the intracellular fluorescence of TRITC-labeled streptomycin in $\Delta hopD$ (2.055 ± 0.397, $P = 0.0019$) and $\Delta hopE$ (1.715 ± 0.602, $p < 0.0001$) mutants was significantly lower than that in the WT strain (2.540 ± 0.597) (Fig. 2A and B), indicating that the two mutants took up significantly less antibiotic than WT strain. Taken together, these results showed that HopE and HopD were responsible for influx of streptomycin and facilitated the antibacterial effect of the antibiotic.

**Natural transformation frequency and rpsLmut synthesis were inhibited by streptomycin after pUC19-*rpsL*mut-*sfgfp* linear plasmid acquisition in *H. pylori* 26695 strain.** Streptomycin is one of the translation-inhibiting antibiotics, which inhibits protein translation by targeting 30S ribosomal protein S12 (*rpsL*) gene. In order to investigate the effect of streptomycin on the antibiotic resistance acquisition by natural transformation in *H. pylori*, we conducted the transformation incubation of WT 26695 strain with StrepR total chromosomal DNA in the absence and presence of streptomycin. Compared to the transformation frequency in the absence of streptomycin ($1.117 \times 10^{-3} \pm 5.455 \times 10^{-4}$), the WT strain displayed more than 6-fold reduction in the yield of recombinant clones ($1.691 \times 10^{-4} \pm 1.536 \times 10^{-4}$) in the presence of streptomycin (5 $\mu$g/mL) (Fig. 3A). This result indicated that the ability of the WT strain

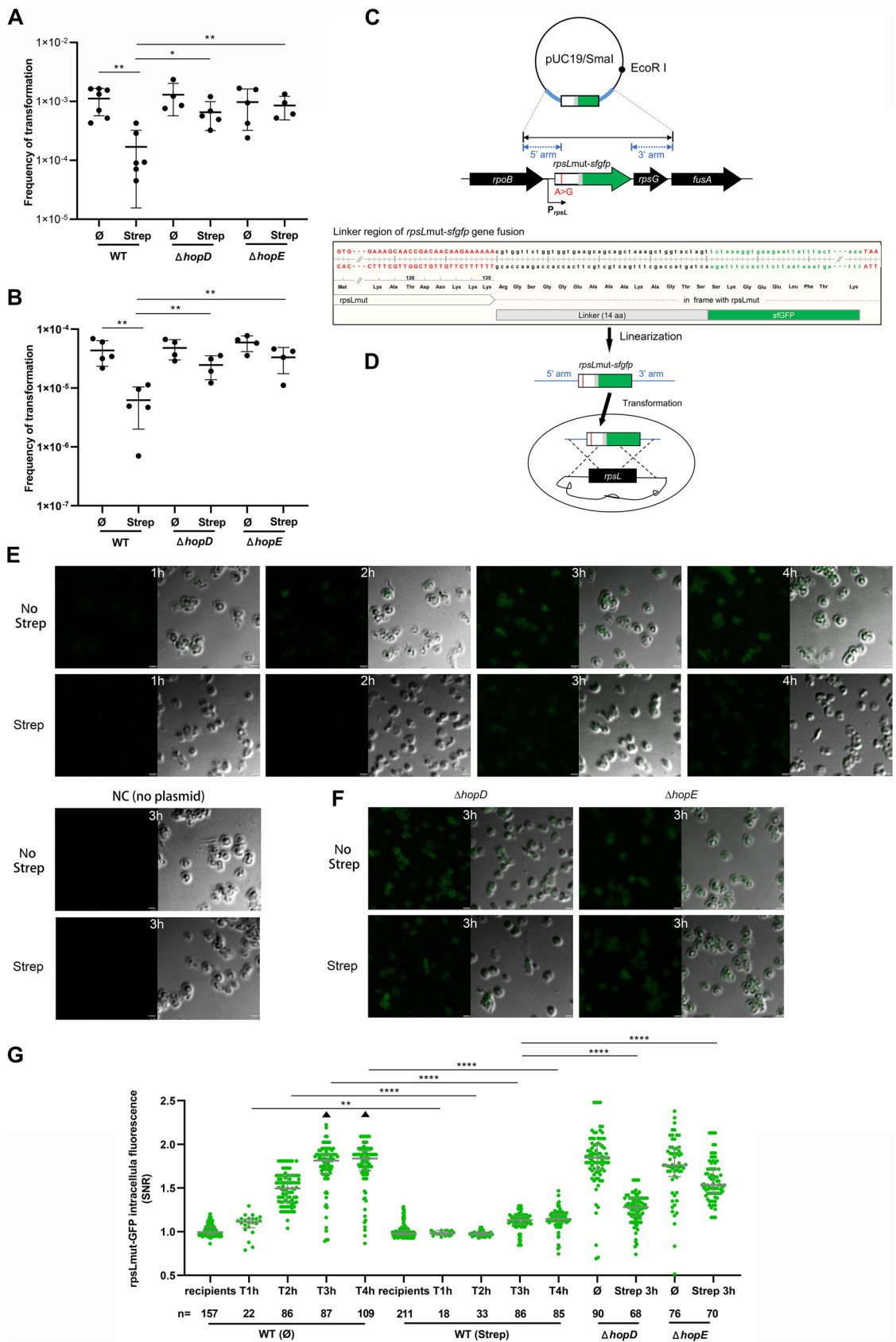

**FIG 3** Recombination frequency and detection of rpsLmut production after the transformation of pUC19-*rpsL*mut-*sfgfp* linear plasmid in the absence and presence of streptomycin in WT, Δ*hopD*, and Δ*hopE* strains. (A, B) Frequency of natural

to convert into resistant transformants was inhibited by the presence of streptomycin. Given that the phenotype conversion of antibiotic resistance requires the expression of resistance-conferring gene, we reasoned that such decreased transformation frequency could be due to the translation-inhibiting effect of streptomycin. To test this hypothesis, we constructed a pUC19-*rpsL*mut-*sfgfp* linear plasmid by ligating *sfgfp* gene with a *rpsL* gene carrying a streptomycin resistance-conferring A128G mutation through linker, which was used to incubate with WT recipient cells and monitor rpsLmut expression (Fig. 3C and D). We compared both the transformation frequency and the rpsLmut production levels. As expected, we observed significant reduction of StrepR colonies, yielding a recombination frequency of $6.205 \times 10^{-6}$ in the presence of streptomycin, compared to the transformation frequency of $4.347 \times 10^{-5}$ in the absence of streptomycin (Fig. 3B). To monitor rpsLmut protein production, we performed microscopy detection of sfGFP at 1, 2, 3, and 4 h after transformation incubation (Fig. 3E), and the intracellular green fluorescence was quantified in recipient and transformant cells. After 1 h, although very few cells expressed rpsLmut-sfGFP, the transformant cells incubated in the presence of streptomycin showed a decrease in fluorescence compared to that of transformants incubated in the absence of streptomycin (Fig. 3E and G). At 3 and 4 h after streptomycin-free incubation, the intracellular fluorescence signal of the transformant cells apparently increased, while only a slight increase of fluorescence signal was observed in transformants incubated with streptomycin (Fig. 3G). The negative control of bacteria incubation without the pUC19-*rpsL*mut-*sfgfp* linear plasmid was performed to show that the *H. pylori* chromosome genome *per se* does not contain the *gfp* gene and that the detected green fluorescent protein (GFP) is completely acquired by transforming (Fig. 3E). These results indicated that the synthesis of the rpsLmut protein after linear plasmid acquisition was inhibited by streptomycin, which failed to resist the antibacterial activity of the antibiotic.

**HopE and HopD deletions restore the resistance acquisition and rpsLmut synthesis by pUC19-*rpsL*mut-*sfgfp* linear plasmid transformation in the presence of streptomycin.** The rpsLmut expression level observed after the acquisition of the plasmid appeared to be affected by the intracellular accumulation of the translation-inhibiting antibiotics. One of the important factors regulating the amount of intracellular antibiotics was the entry rate of antibiotic molecules into the recipient cells. Therefore, we assumed that HopE and HopD porins responsible for streptomycin influx may affect the rpsLmut expression and phenotypic transformation frequency. To determine the contribution of HopE- and HopD-mediated influx, respectively, we evaluated the transformation frequency and quantified the rpsLmut production after transfer of the linear plasmid in the absence and presence of streptomycin in Δ*hopD* and Δ*hopE* mutants. Interestingly, we found that, for the isogenic StrepR DNA transformation, the mutant recipient cells were still able to acquire a degree of resistance in the presence of streptomycin, yielding a resistance acquisition frequency of $6.560 \times 10^{-4}$ (Δ*hopD* mutants)

**FIG 3** Legend (Continued)
transformation of StrepR total chromosomal DNA and *rpsL*mut-*sfgfp* linear plasmid (the frequency of recipient cells converted into streptomycin-resistant transformants) in WT, Δ*hopD*, and Δ*hopE* strains, estimated by plating assay 8 h after mixing the DNA with recipient cells, in the absence (Ø) and in the presence of streptomycin (5 μg/mL), with the mean and standard deviation calculated from at least four biological repeats (black dots). (C) Diagram of pUC19-*rpsL*mut-*sfgfp* plasmid and sequence of the linker region between *rpsL*mut and *sfgfp* genes. The *sfgfp* gene was in frame with *rpsL*mut sharing the start codon and stop codon. (D) Diagram of the pUC19-*rpsL*mut-*sfgfp* transformation experiment by homologous recombination. Bacteria were incubated with the *rpsL*mut-*sfgfp* linear fragment containing the flanking sequences. The production of rpsLmut could be then detected by expression of fluorescent sfGFP. (E) Representatives of microscope sfGFP detection in the WT strain at 1, 2, 3, and 4 h after incubation in the absence and presence of streptomycin. The negative control (NC) of bacteria incubation after 3 h without the pUC19-*rpsL*mut-*sfgfp* linear plasmid was also displayed. (F) Representatives of microscope sfGFP detection in Δ*hopD* and Δ*hopE* mutants 3 h after incubation in the absence and presence of streptomycin. Automatic green fluorescent protein (GFP) channel and merged images of GFP and DIC are shown. The scale bars correspond to 1.5 μm. (G) Scatterplots showing the single-cell quantification of intracellular fluorescence of sfGFP (SNR) in WT, Δ*hopD*, and Δ*hopE* strains at the indicated times after incubation of linear pUC19-*rpsL*mut-*sfgfp* with bacteria, in the absence (Ø) and presence (Strep) of streptomycin (5 μg/mL). The median, quartile 1, and quartile 3 are indicated by horizontal lines. Black dots above and below the maximum and minimum values correspond to outlier cells. Triangles indicate the presence of data points beyond the axis range. The numbers of cells analyzed (n) are also indicated. (A, B, and G). *, significant ($P < 0.05$); **, very significant ($P < 0.01$); ****, highly significant ($P < 0.0001$). The source data are provided as source data files. aa, amino acids.

and $8.532 \times 10^{-4}$ ($\Delta hopE$ mutants), respectively (Fig. 3A), which was significantly higher than that of the WT strain with the transformation frequency of $1.691 \times 10^{-4}$ in the presence of streptomycin ($\Delta hopD$ mutants: $P = 0.011$; $\Delta hopE$ mutants: $P = 0.003$). Similarly, for the pUC19-*rpsL*mut-*sfgfp* linear plasmid transformation, the mutant recipient cells still managed to acquire resistance in the presence of streptomycin, yielding a recombination frequency of $2.457 \times 10^{-5}$ ($\Delta hopD$ mutants) and $3.319 \times 10^{-5}$ ($\Delta hopE$ mutants), respectively (Fig. 3B), which was significantly higher than that of the WT strain with the transformation frequency of $6.205 \times 10^{-6}$ in the presence of streptomycin ($\Delta hopD$ mutants: $P = 0.009$; $\Delta hopE$ mutants: $P = 0.007$). To show specificity, three other porin-deleted mutants were also used to measure the transformation frequency, which showed that, in the presence of streptomycin, the $\Delta hopA$, $\Delta hopB$, and $\Delta hopC$ mutants all showed unchanged resistance acquisition compared with that in the WT strain when incubated with neither isogenic StrepR DNA nor pUC19-*rpsL*mut-*sfgfp* linear plasmid (Fig. S2). We also performed microscopy evaluation of rpsLmut production in $\Delta hopD$ and $\Delta hopE$ mutant transformants after transformation with the pUC19-*rpsL*mut-*sfgfp* linear plasmid (Fig. 3F and G). We observed that although the intracellular fluorescence signals in the two mutant transformants were affected by streptomycin treatment, both revealed apparently higher rpsLmut production levels (especially $\Delta hopE$ mutant transformants) compared to that of WT transformants in the presence of streptomycin (Fig. 3G) (both $\Delta hopD$ and $\Delta hopE$ mutant transformants: $P < 0.0001$). Thus, rpsLmut production was enhanced after plasmid acquisition in the presence of streptomycin when HopD and especially HopE porins were deleted, respectively. The ability of $\Delta hopE$ mutant to restore proteins despite the presence of streptomycin was further confirmed by monitoring the production of sfGFP after the acquisition of the pUC19-$P_{ureA}$sfgfp plasmid carrying the *sfgfp* gene under the control of the $P_{ureA}$ (promoter of *ureA* gene) constitutive promoter (Fig. S3). The result showed that reduced sfGFP expression in the WT strain in streptomycin treatment was significantly restored in $\Delta hopE$ mutant (Fig. S3A and B). In addition, we excluded the effect of streptomycin treatment on the viability of the WT, $\Delta hopD$, and $\Delta hopE$ strains during acquiring the linear plasmid by showing that transient exposure (up to 8 h) to streptomycin did not affect the survival of the strains measured by plating assays (Fig. S4).

**HopE deletion alleviates the inhibition of protein synthesis by streptomycin.** To test whether HopE deletion could mitigate the inhibition effect of streptomycin on protein synthesis in terms of the whole proteome in *H. pylori*, we performed a reversed-phase liquid chromatography-mass spectrometry (RPLC-MS) analysis of total proteins in WT and $\Delta hopE$ strains before and after streptomycin treatment. A total of 878 protein were detected in the WT strains, and we found that streptomycin treatment reduces the relative abundance of 7.97% (70 of 878) of the detected proteins in WT cells (Fig. 4A; Table S6). In $\Delta hopE$ mutants, this percentage significantly reduced to 3.14% (16 of 859, $P < 0.0001$) (Fig. 4B; Table S7), corroborating that HopE deletion attenuated the inhibition of protein synthesis in the presence of streptomycin.

**Resistance acquisition frequency of WT, $\Delta hopD$, and $\Delta hopE$ strains after transformation in the presence of other antibiotics.** Finally, we tested the role of HopD and HopE deletions for the acquisition of streptomycin resistance in the presence of other three protein synthesis-inhibiting antibiotics clarithromycin, tetracycline, and rifampin and a DNA-damaging agent metronidazole. These experiments revealed that HopE deletion greatly contributed to the acquisition of streptomycin resistance in the presence of rifampin (inhibits transcription) and metronidazole (damages DNA) but showed no effect in the presence of clarithromycin (inhibits translation) and tetracycline (also inhibits translation). $\Delta hopD$ mutant retained a slight but not significant increase in the transformation frequencies compared to those of the WT strain in the presence of all four antibiotics (Fig. 5).

## DISCUSSION

Antibiotic uptake and efflux by *H. pylori* are the basis for intrinsic susceptibility to antibiotics (25). The permeability barrier in Gram-negative pathogens, including porin-mediated passive uptake across the outer membrane and active efflux via efflux pumps in the inner

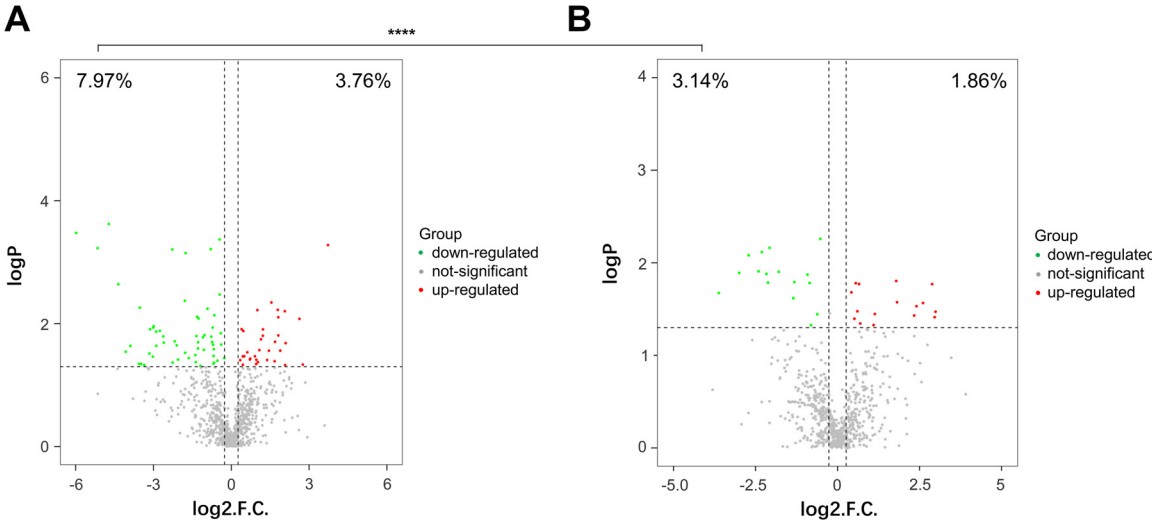

**FIG 4** Proteomic analysis showing relative abundance alterations of proteins in WT and Δ*hopE* strains before and after streptomycin treatment. The volcano plots show the relative abundance alteration in WT and Δ*hopE* strains treated with streptomycin (5 μg/mL) compared to that in the two strains not treated with streptomycin. Fold change (FC) analysis and *t* tests were used to perform univariate statistical analysis on relative abundance of the proteins. Proteins with fold change greater than or less than 1.2 and significance analysis *P* value less than 0.05 were selected as significantly different proteins. Green spots were abundance-downregulated proteins, red spots were abundance-upregulated proteins, and gray spots were not significant proteins. (A) WT strains. (B) Δ*hopE* strains. The percentages of downregulated and upregulated proteins were annotated in the figure. ****, highly significant (*P* < 0.0001; chi-square test).

membrane, contributes to the emergence of antibiotic resistance. Understanding porin-mediated entry of substrates is important owing to its key role in defining antibiotic pathways to their targets (26). From the perspective of porin-mediated influx in *H. pylori*, our study reveals the important role of HopE and HopD porins in not only preserving the intrinsic susceptibility to specific antibiotic but also evading acquired antibiotic resistance by NT in the presence of translation-inhibiting antimicrobial.

The MIC values of all tested antibiotics on the mutants lacking HopE were significantly increased, which was consistent with a previous study using model planar lipid membrane systems showing that HopE porin acts principally as a nonspecific channel (21). Dramatically, the mutant lacking HopD also showed increased MIC values to partial antibiotics, including tetracycline, streptomycin, and levofloxacin. Similarly, the

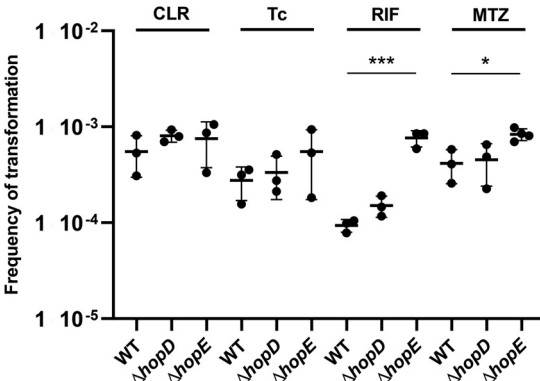

**FIG 5** Streptomycin resistance acquisition frequencies of WT, Δ*hopD*, and Δ*hopE* strains after transformation in the presence of other antibiotics. Frequency of natural transformation of StrepR total chromosomal DNA in WT, Δ*hopD*, and Δ*hopE* strains, estimated by plating assay 8 h after mixing the DNA with corresponding recipient cells, in the presence of clarithromycin (CLR, 0.5 μg/mL), tetracycline (Tc, 0.1 μg/mL), rifampin (RIF, 0.1 μg/mL), and metronidazole (MTZ, 4 μg/mL), respectively. Means and standard deviation were calculated from at least three biological repeats (black dots). *, significant (*P* < 0.05); ***, highly significant (*P* < 0.001). The source data are provided as source data files.

subsequent growth curve experiment showed that the 80-h maximal growth of the ΔhopE and ΔhopD mutants was enhanced in the presence of subinhibitory concentrations of all or some of the antibiotics tested. Notably, when the environmental antibiotics were beyond the subinhibitory concentrations, the abilities of the two mutants to resist the antibacterial effect of the drugs were undermined and even destroyed. The observation may be the product of the continuing substantial drug penetration through a lipid-mediated pathway depending on the hydrophilic or hydrophobic property of the antibiotic molecules (26) rather than the general diffusion through porins and high drug concentrations beyond the upper limit of antibiotic efflux levels mediated by efflux pump. Moreover, it has been recently demonstrated that multiple intracellular activities regulate bacterial membrane permeability in response to external pressure (26). Therefore, further interpretation of the specific function and structure of effective porins related to substrate uptake would facilitate our understanding of the molecular basis of antibiotic uptake through porins, which takes a place in the complicated antibiotic resistance mechanism (27–30).

HGT is a main contributor to the spread of antibiotic resistance genes (3). Conversely, it was generally assumed that antibiotics can selectively drive the HGT of ARGs and form the promotive effects (31–34), which mainly focused on conjugation-induced horizontal acquisition of resistance due to its extensive prevalence. Existing research seemed to explain this conclusion from different perspectives. It was reported that application of antibiotics has likely influenced conjugation-induced HGT by exerting selection that increased the proportion of donor cells offering resistance coded by mobile genetic elements (MGEs) (35). In addition, bacteria with higher transfer efficiencies might be selected by antibiotic treatment, which potentially regulated the HGT rate (36). Another study showed that plasmid conjugation in *E. coli* was boosted with a combination of kanamycin and streptomycin (37). Nonetheless, reassessing these results indicated that it was the population expansion over time of the transconjugants that was reported rather than the proposed elevated conjugation efficiency (31). In addition, there was almost no difference in the conjugation results in the presence and absence of antibiotics in the first several hours of conjugation (31). In addition, this kind of HGT can be stimulated by antibiotics through inducting the bacterial DNA damage (SOS) response, a widespread system in bacteria that promotes cell survival by activating a set of coregulated genes in response to DNA damage (38). Beaber et al. (39) reported that the activated SOS DNA damage response in *Vibrio cholerae* donor cells was associated with higher transfer efficiency of resistance coded by a integrative conjugative element (ICE) by relieving the repression effect of a regulator SetR on the transfer ability of the ICE. This finding has implications for the use of antibiotics because many antibiotics at nonlethal concentrations can trigger the SOS response and enhance resistance dissemination. Moreover, in a convincing series of experiments, the transfer and integrase functions of an ICE were activated in *E. coli* and *V. cholerae* by two DNA-damaging antibiotics (mitomycin C and ciprofloxacin), which required the coprotease activity of RecA, a recombinase for DNA repair (39, 40). Similarly, some other studies have regarded increased frequency as a product of indirect induction of conjugation by antibiotics as well (37, 41, 42). However, as to *H. pylori*, it has been demonstrated that this pathogen was distinct in DNA damage response because it triggers both DNA uptake machinery and an enzyme that liberates DNA from neighboring cells rather than activating SOS DNA repair systems. This capacity for genetic exchange enhances recombination of exogenous DNA into the genome, thus contributing to the spread of antibiotic resistance (43). Similar views have also been drawn from several studies indicating that the very presence of antibiotics, especially those affecting DNA replication or causing DNA damage, may facilitate NT-induced HGT and the dissemination of resistance genes in certain bacteria by eliciting a stress response that enhances expression of competence genes or by increasing the fraction of competent cells in laboratory conditions (44–46). For instance, both the expressions of competence genes and transformation rates were significantly enhanced in *Streptococcus* after quinolones treatment (47). Moreover, replication stalling induced by stress in *S.*

*pneumoniae* can lead to increased transcription of the competence gene cluster (46). However, other antibiotics did not have such an effect.

Instead of investigating the transformation efficiency affected by alterations of DNA uptake and integration caused by DNA-damaging antibiotic utilization through measuring the expression level of competence proteins in *H. pylori*, which were not within the scope of this study, we chose another perspective, focusing on the effect of cellular accumulation of antibiotics, especially protein synthesis-inhibiting antibiotics, controlled by putative porins, on resistance acquisition via directly detecting the NT frequency and acquired resistance gene expression. In this study, we differentially found that NT frequency of the WT strains displayed more than 6-fold reduction in the presence of streptomycin. Here, we speculated the possible explanation could be the inhibited expression of the transforming resistance gene by the translation-inhibiting effect of streptomycin, which was demonstrated in our consequent experiments. Several other studies on conjugation also showed results inconsistent with the previously established view that antibiotics promote HGT (32, 48, 49). Ciprofloxacin can prevent R-plasmid transfer by inhibiting conjugation in *E. coli* (50). Also, sub-MIC of mupirocin diminished the conjugative transfer of the gentamicin resistance plasmid pWG613 among *Staphylococcus aureus* strains by more than 1,000-fold. Most importantly, in 2019, Nolivos et al. (48) demonstrated that conjugation frequency was reduced to nearly 50% by translation-inhibiting antibiotics, while there was still a majority of transconjugants acquiring resistance, which was a benefit of the AcrAB TolC efflux activity alleviating the inhibitory effect of drugs on the reactions that are required for resistance establishment after plasmid transfer. This also did not completely coincide with our result that translation-inhibiting antibiotic treatment led to greater reduction in resistance acquisition in this study, and we speculated that the possible reason may be the inefficient efflux of streptomycin by the main AcrAB TolC efflux pump in *H. pylori*.

We attempted to investigate how the transformation frequency was inhibited. Taken together with our further evaluation of the synthesis of transformed resistance gene by constructing a pUC19-*rpsL*mut-*sfgfp* linear plasmid carrying a streptomycin resistance-conferring mutation, the results implicated that the declined resistance acquisition may be at least partially due to the inhibited rpsLmut synthesis by streptomycin. Therefore, it was likely that reduced expression of *rpsL* carrying a resistance-conferring mutation acquired by transformation in the presence of streptomycin incapacitated WT cells to produce rpsL protein with reduced binding affinity with streptomycin, thus leading to the failure of increasing the chance of evading streptomycin antibacterial activity, which eventually led to the death of cells. This likely negative feedback cycle also interestingly showed that the *rpsL* gene may be within the gene substrates whose expression can be regulated by *rpsL* itself. However, this effect mode could be reversed by HopE or HopD deletion in our study. By observing the reduced streptomycin accumulation in mutants lacking HopE or HopD, we assumed that HopE or HopD porin affected antibiotic acquisition via controlling antibiotic uptake, which influenced the resistance-conferring gene expression and thus inhibited that negative feedback in *H. pylori*. Further proteomic analysis also substantiated this view by revealing that HopE deletion attenuated the inhibition of chromosomal protein synthesis levels in the presence of streptomycin. Of note, among all the detected proteins, *rpsL* indeed downregulated but did not reach a significant decrease. This disparity could be likely explained by the primary limitations in label-free proteomics that it has difficulties in ensuring the quantification reproducibility and robustness and has obstacles to achieving its best sensitivity (51, 52). Moreover, compared with the high-throughput proteomics analysis technology, quantification of the expression specific to this protein seems more confirming.

To answer the question regarding whether streptomycin may inhibit the transforming DNA expression by regulating the NT machinery proteins, the explanation could be as followed: it seems not time-allowed for NT machinery proteins to make possible responses before transformation occurs. During the NT incubation of strains with transforming DNA, streptomycin was added at the same time with the transformation mix.

However, all of the transformation steps from DNA seizing, recombination, and integration to expression, occur in less than 1 h, and the DNA is fully expressed in 2 to 3 h (10), relying on the exiting levels of NT machinery proteins regardless of whether streptomycin affected expression of these proteins because they also need time to respond and alter their expressions or not.

In the presence of the transcription-inhibiting antibiotic rifampin and the DNA-damaging agent metronidazole, the streptomycin resistance acquisition frequencies also maintained apparently higher levels in mutants lacking HopE but not HopD after StrepR DNA transformation compared to those in the WT strain, which showed that the loss of HopE plays a more important role in resistance dissemination in the presence of indicated concentrations of antibiotics. Metronidazole is one of the first-line antibiotics for *H. pylori* eradication with a low molecular weight of 171.2, whose metabolite can block bacterial nucleic acid synthesis (53). Hence, our results implied that HopE loss could also preserve resistance acquisition by transformation in the presence of antibiotics with other modes of action. As to StrepR DNA transformation in the presence of clarithromycin and tetracycline in our final experiment, which are classic translation-inhibiting antibiotics, we did not observe the significantly higher resistance acquisition in mutants compared to the same high levels of those in the WT strain. Taken together, our findings could be explained as follows. The acquired gene expression observed after transformation was bound to be affected by the intracellular accumulation of the translation- or transcription-inhibiting antibiotics such as streptomycin and rifampicin. Therefore, in the presence of these antibiotics, it is possible that sufficient drug accumulation in the WT strain and reduced drug concentration in mutant cells caused the significant difference in NT frequency (and protein expression). While in the presence of clarithromycin or tetracycline, the unobserved mechanism that *hopE* deletion did not significantly restore NT frequency was actually due to the same high level of NT frequency in the WT strain, which could be due to the insufficient drug accumulation in the WT strain. It is known that the intracellular levels of antibiotics are regulated by the balance of two main factors: porin-mediated influx and efflux pump-mediated efflux, which have their own substrates (54, 55). Therefore, it could be speculated that the sufficient accumulation of streptomycin and rifampicin in the WT strain may be due to the lack of efficient efflux of these antibiotics, and the insufficient accumulation of clarithromycin or tetracycline in the WT strain was probably due to the efficient efflux of the two antibiotics by efflux pump in *H. pylori*. Perhaps clarithromycin and tetracycline rather than rifampicin and metronidazole were within the substrates of the main efflux pump in *H. pylori*.

**Conclusion.** Our study reveals the important role of HopE, and to a minor extent HopD porins, not only in preserving the intrinsic susceptibility to specific antibiotic but also in evading drug-specific resistance acquired by NT in the presence of bacterial inhibitors. The loss of HopE and/or HopD porins in *H. pylori* genomes, combined with the large number of secreted or cell-free genetic elements carrying single nucleotide mutations conferring *H. pylori* antibiotic resistance, increases the probability that this pattern serves as an important mechanism for the spread of drug resistance within bacterial communities.

## MATERIALS AND METHODS

***H. pylori* culture conditions.** The cultures of *H. pylori* 26695 (wild-type [WT]) and mutants were grown under microaerophilic conditions (5% $O_2$, 10% $CO_2$, and 85% $N_2$, using the MAC-MIC system from AES Chemunex or an airtight culture container with the Mitsubishi Gas Company anaeropack) at 37°C. Karmali (Oxoid) medium supplemented with 10% defibrinated horse blood was used for plate cultures. Liquid cultures were grown in *Brucella* broth (Becton Dickinson, Sparks, MD) supplemented with 10% decomplemented fetal bovine serum (FBS) (Invitrogen, Carlsbad, CA) with constant shaking (150 rpm). Antibiotic mix containing polymyxin B (0.155 mg/mL), vancomycin (6.25 mg/mL), trimethoprim (3.125 mg/mL), and amphotericin B (1.25 mg/mL) was added to both plate and liquid cultures. An UV spectrophotometer was used to detect bacterial concentration (1.0 $A_{600}$ = 1 × $10^8$ CFU/mL).

**Construction of gene variants.** *H. pylori* 26695 was used as parental strain to generate all the gene variants. The gene sequences were obtained from the annotated complete genome sequence of *H. pylori* 26695 deposited at Pylori Gene World-Wide Web Server (http://genolist.pasteur.fr/PyloriGene/). For chromosomal integrations/replacements of gene variants, the specific flanking sequences (Table S2)

wherever appropriate for *hopA* (556 bp for 5′ arm and 511 bp for 3′ arm), *hopB* (510 bp for 5′ arm and 567 bp for 3′ arm), *hopC* (532 bp for 5′ arm and 533 bp for 3′ arm), *hopD* (511 bp for 5′ arm and 510 bp for 3′ arm), and *hopE* (573 bp for 5′ arm and 578 bp for 3′ arm) gene regions were amplified from genomic DNA (gDNA) of the *H. pylori* 26695 strain using high-fidelity PCR enzyme and sequence-specific primers (Table S3), were combined by PCR with a kanamycin resistance cassette amplified from pEGFP-N1 or pET28a plasmid (using primers KnF and KnR; Table S3) in the middle, and were cloned to the pUCmT vector to separately generate the recombinant plasmids pUCmT-Δ*hopA*::Kn, pUCmT-Δ*hopB*::Kn, pUCmT-Δ*hopC*::Kn, pUCmT-Δ*hopD*::Kn, and pUCmT-Δ*hopE*::Kn (Table S4). All the plasmids harboring different gene constructs were introduced into *H. pylori* 26695 by electroporation transformation for allelic replacement of native *hopA*, *hopB*, *hopC*, *hopD*, and *hopE* locus, respectively. Transformants were selected by using plates containing kanamycin (20 $\mu$g/mL). Partial positive clones were inoculated into liquid cultures. The correct constructions were verified by PCR using 5′- and 3′-flanking sequence (homologous arms) primers and gene-specific primers (Table S3). The glycerol stocks of the *H. pylori* strains were prepared in *Brucella* broth media supplemented with 20% glycerol and stored in −80°C.

**Determination of antibiotic susceptibility.** The antimicrobial resistance of *H. pylori* 26695 and mutants to metronidazole (MTZ), clarithromycin (CLR), levofloxacin (LEV), amoxicillin (AMX), and tetracycline (Tc) and rifampin (RIF) were determined by ETEST (Kangtai Biotechnology Co., Ltd.; Autobio, China) and by the agar dilution method. In the ETEST experiment, chloramphenicol not affected by all mutants was used to show specificity. The antibiotic susceptibility of *H. pylori* 26695 and mutants to streptomycin (Strep) was determined by an agar dilution method. All antimicrobials were purchased from Wako Pure Chemical Industry (Osaka, Japan). For the ETEST, the culture suspension turbidity of *H. pylori* was adjusted with saline to a McFarland opacity standard of 2.0, and the suspensions were inoculated onto an MH agar plate (Becton Dickinson, Sparks, MD) supplemented with 10% horse blood. The ETEST strips for the six drugs were inserted into the plate and incubated at 37°C for 3 to 5 days under microaerophilic conditions. The agar dilution tests were performed according to the protocols of the Clinical and Laboratory Standards Institute (CLSI, Wayne, PA). Briefly, each strain was suspended in sterile saline at a density equivalent to a McFarland opacity standard of 2.0 to 3.0, and each suspension (1 mL) was spotted onto MH agar plates supplemented with 5% sheep blood containing 2-fold serial dilutions (ranging from 0.001 to 32 $\mu$g/mL) of each antibiotic. An antibiotic-free plate was inoculated before and after each series of antibiotic plates to confirm viability of the inoculum and observe possible contamination. The plates were incubated under microaerobic conditions at 37°C for 3 days. The MIC was determined as the lowest concentration of antibiotic that apparently decreased the growth relative to the control. *H. pylori* ATCC 43504 was used as the quality control reference strain. The agar dilution tests were performed a minimum of three separate times.

**Constructions of translational fusions.** Chemically synthetic 2,128-bp double-stranded DNA (dsDNA) containing hp1197 gene carrying A128G mutation and 500+-bp flanking sequences and a *sfgfp*-linker sequence (*rpsL*mut-*sfgfp*) (Table S5) was cloned into pUC19/Smal vector to generate the pUC19-*rpsL*mut-*sfgfp* plasmid, which was transformed into *E. coli* TOP10. The plasmid was purified using UNIQ-500 column plasmid Max-Preps kit (Sangon Biotech, Shanghai, China) and linearized by EcoRI (TaKaRa,15 U/$\mu$L), which was used as a substrate for the NT experiment.

**Growth curves and spot assay.** Growth curves were performed automatically using TECAN Spark multimode plate reader. WT, Δ*hopA*, Δ*hopB*, Δ*hopC*, Δ*hopD*, and Δ*hopE H. pylori* cells were inoculated into *Brucella* broth (1:100) containing 10% decomplemented FBS in the presence or absence of Strep (1 and 5 $\mu$g/mL), MTZ (1 and 4 $\mu$g/mL), CLR (0.05 and 0.5 $\mu$g/mL), LEV (0.01 and 0.1 $\mu$g/mL), AMX (0.1 and 0.5 $\mu$g/mL), Tc (0.1 and 1 $\mu$g/mL), and RIF (0.1 and 1 $\mu$g/mL). Then transparent, flat-bottomed 96-well plates were loaded with 200 $\mu$l of the *Brucella* broth cultures and incubated under microaerophilic conditions at 35°C, with constant shaking and measuring optical density at 600 nm (OD$_{600}$) every 5 h over 80 h. Mean growth curves with standard deviation were generated using GraphPad Prism. For transient exposure to streptomycin, WT, Δ*hopD*, and Δ*hopE* strains were inoculated (1:100) in *Brucella* broth containing 10% decomplemented FBS and were grown to an OD$_{600}$ of 4.0. Streptomycin (5 $\mu$g/mL) was then added to the culture, which was then incubated under microaerobic conditions at 37°C with shaking. After 4 and 8 h, a sample of the culture was serial diluted, and 10 $\mu$l drops were deposited on Karmali plates. The sample was then cultured for 3 days. The plates were imaged using a Bio-Rad ChemiDoc MP system via a customization mode with white epi illumination.

**Natural transformation assay.** To obtain the isogenic streptomycin resistance (StrepR) total chromosomal DNA, exponentially growing *H. pylori* 26695 cells from liquid cultures (OD$_{600}$ of 4.0) were plated on streptomycin-containing (5 $\mu$g/mL) plates and incubated for 5 to 7 days. The frequency of StrepR colonies is on the order of $10^{-9}$. After three subcultures, the total genomic DNA of the StrepR cells was extracted using DNAzol (Thermo Fisher Scientific, Waltham, MA) and kept at −80°C. All the procedures were performed following the manufacturer's instructions. A total of 500 ng of this genomic DNA or *rpsL*mut-*sfgfp* linear plasmid DNA with 60 $\mu$l of exponentially growing *H. pylori* 26695 or mutant cells from liquid cultures resuspended in 1 mL *Brucella* broth containing 10% FBS were mixed into the wells of 24-well plates and incubated for 8 h at 37°C without shaking. The cells were washed by centrifugation (5,000 rpm/min); resuspended by peptone water; then vortexed, serial diluted, and plated on Karmali plates with or without streptomycin (5 $\mu$g/mL); and incubated for 4 to 5 days at 37°C. The transformation frequencies were calculated as the number of StrepR colonies/recipient CFU. When NT was done in the presence of antibiotics (Strep, MTZ, Tc, CLR, and Rif), supplements were added at the same time as the transformation mix and maintained all along the experiment.

**Fluorescence microscopy experiments and image analysis.** To determine the in vivo *rpsL*mut-*sfgfp* expression level, 500 ng of the linear plasmid and 60 $\mu$l of exponentially growing cells from liquid cultures

resuspended in 1 mL *Brucella* broth containing 10% FBS were mixed into wells of 24-well plates in the absence or in the presence of streptomycin (5 $\mu$g/mL) and incubated for 1, 2, 3, and 4 h or indicated time at 37°C without shaking. The cells were washed twice with phosphate-buffered saline (PBS) and were resuspended in 500 $\mu$l of sterile saline and spotted on coated gelatin glass coverslips and fixed for 1 h with 4% formaldehyde. The slides were washed three times with PBS and mounted in a SlowFade diamond mountant (S36963, Thermo Fisher Scientific, Waltham, MA). Image acquisition was performed with a Leica TCS SP8 STED confocal microscope (Wetzlar, Germany) using a 100×/1.40 oil lens. Acquisition settings included an automatic GFP channel (488-nm absorption laser [10% transmission] and 498- to 577-nm emission laser) and a differential interference contrast (DIC) detector. The image processing was performed and the signal/noise ratio (SNR, a proxy for the relative concentration of fluorescence inside the cells) was determined manually using NIS-element software (Nikon Corp., Tokyo, Japan). SNR here corresponds to the ratio (intracellular signal/noise signal) at which the intracellular signal is the fluorescence signal per cell area and the noise is the signal measured outside the cells (due to the fluorescence emitted by the surrounding medium). For streptomycin accumulation analysis, exponentially growing cells from liquid cultures resuspended in 1 mL *Brucella* broth containing 10% FBS were incubated with 10 $\mu$g/mL TRITC-labeled streptomycin (Xi'an Qiyue Biotechnology Co., Ltd., China) for 1 h at 37°C without shaking. Cell wash and fix and image acquisition were as described above. Acquisition settings included an automatic red fluorescence channel and a DIC detector. Image processing and single-cell level SNR calculation were done manually the same as described above.

**Reversed-phase liquid chromatography-mass spectrometry (RPLC-MS) analysis.** The samples were analyzed in triplicate using a Dionex Ultimate 3000 RSLCnano (Thermo Scientific, San Jose, CA) with chromatographic column (75-$\mu$m inner diameter × 150-mm, packed with Acclaim PepMap RSLC C18, 2 $\mu$m, 100 Å, nanoViper) coupled on line with a Thermo Scientific Q Exactive PLUS mass spectrometer (Thermo Scientific, San Jose, CA). We analyzed one quadruplex containing WT with no Strep, WT with Strep for 3 h, $\Delta hopE$ with no Strep, and $\Delta hopE$ with Strep for 3 h. The samples were centrifuged at 8,000 × *g* at 4°C and added 1 mL precold RIPA lysis solution containing 5 $\mu$L (0.5%) phosphatase inhibitor, 1 $\mu$L (0.1%) protease inhibitor, and 10 $\mu$L (1%) phenylmethylsulfonyl fluoride (PMSF, a protease inhibitor), which were subjected to sonication eight times (50W power) in protein dissolution buffer. The obtained proteins subsequently underwent quality inspection by quantification using a BCA quantitative kit (23275, Thermo Scientific, San Jose, CA) and SDS-PAGE electrophoresis and then underwent protease digestion. The peptide fractions were dissolved in 20 $\mu$L solution (0.1% formic acid and 5% acetonitrile), fully oscillated, and centrifuged at 13,500 rpm at 4°C for 20 min. The supernatants were transferred to the loading tube, and 8 $\mu$L was absorbed for mass spectrometry identification. The parameter for mobile phase A was 0.1% formic acid (64186, Thermo Scientific), and that for mobile phase B was 0.1% formic acid and 80% ACN (A998-4, Thermo Fisher Scientific). The separated peptides directly entered the mass spectrometer Thermo Scientific Q Exactive Plus for online detection. Maxquant software (version 1.6.17.0) was used for peptide retrieval in the UniProt proteome database (UP000000429_20211001.fasta). Fold change (FC) analysis and *t* test were used to perform univariate statistical analysis on the proteins. Proteins with fold change greater than or less than 1.2 and with a significance analysis *P* value less than 0.05 were selected as proteins with significantly different abundance, and then the volcano plots were generated.

**Statistical analysis.** Differences between groups were compared using the MWU Mann-Whitney U test, Student's *t* test, one-way analysis of variance (ANOVA), and chi-square test according to the data type and the number of comparisons. When the results of one-way ANOVA tests showed a statistical difference, the intragroup pairwise comparison was performed using Dunnett's test or Tukey's test. The statistical analysis was performed via GraphPad Prism (version 8.0). A *P* value less than 0.05 was considered statistically significant.

**Data availability.** The main data supporting the findings of this study are available within the article and its supplementary figures and tables. The source data for the figures are provided as a source data file.

## SUPPLEMENTAL MATERIAL

Supplemental material is available online only.
**SUPPLEMENTAL FILE 1**, PDF file, 0.6 MB.
**SUPPLEMENTAL FILE 2**, XLSX file, 0.02 MB.
**SUPPLEMENTAL FILE 3**, XLSX file, 0.01 MB.
**SUPPLEMENTAL FILE 4**, XLSX file, 0.02 MB.
**SUPPLEMENTAL FILE 5**, XLSX file, 0.01 MB.
**SUPPLEMENTAL FILE 6**, XLSX file, 0.9 MB.
**SUPPLEMENTAL FILE 7**, XLSX file, 0.9 MB.
**SUPPLEMENTAL FILE 8**, XLSX file, 1.3 MB.

## ACKNOWLEDGMENTS

This work was supported by Shanghai Science and Technology Committee grants 18411960600, 16411968000, and 18411950800; National High Technology Research and Development Program of China grant 2015AA021107-019; Shanghai Shenkang Hospital Development Center New Frontier Technology Joint Research Project grant SHDC12015107; Shanghai Health and Family Planning Commission Youth Project of

Scientific Research Subject grant 20184Y0032; funds from the Shanghai Rising Stars of Medical Talent Outstanding Youth Development Program of 2018; and Huadong Hospital Project grant 2019jc019.

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
