## [Reviewer comments · Microbiology Spectrum]

Microbiology Spectrum

HopE and HopD porins-mediated drug influx contribute to the intrinsic antimicrobial susceptibility and inhibit streptomycin resistance acquisition by natural transformation in *Helicobacter pylori*

Yixin Liu, Feng Yang, Su Wang, Wenjing Chi, Li Ding, Tao Liu, Feng Zhu, Dalian Ji, Jun Zhou, Yi Fang, Jinghao Zhang, Ping Xiang, Yanmei Zhang, and Hu Zhao

Corresponding Author(s): Yanmei Zhang, Huadong Hospital affiliated to Fudan University

Review Timeline:

Submission Date:	October 20, 2021
Editorial Decision:	November 10, 2021
Revision Received:	December 11, 2021
Editorial Decision:	December 22, 2021
Revision Received:	January 17, 2022
Accepted:	January 21, 2022

Editor: Beile Gao

Reviewer(s): Disclosure of reviewer identity is with reference to reviewer comments included in decision letter(s). The following individual involved in review of your submission agreed to reveal their identity: Santanu Chattopadhyay (Reviewer #2)

Transaction Report:

DOI: <https://doi.org/10.1128/Spectrum.01987-21>

November 10, 2021

Prof. Hu Zhao
Huadong Hospital affiliated to Fudan University
Shanghai
China

Re: Spectrum01987-21 (The effect of porins in the intrinsic antimicrobial tolerance and resistance acquisition by natural transformation in *Helicobacter pylori*)

Dear Prof. Hu Zhao:

Link Not Available

Sincerely,

Beile Gao

Journals Department
Reviewer comments:

Reviewer #1 (Comments for the Author):

In this manuscript, Liu et al. explore the consequences of inactivating porins on the antibiotic susceptibility and the natural transformation capacity of *H. pylori*. For that, they generated mutant strains defective in each of five different porin genes (hopA-E) and determined their MIC for 7 different antibiotics. They also monitored the capacity of the mutants to uptake a fluorescent analogue of streptomycin. From these experiments the authors concluded that porins HopE and D absence leads to a reduced influx of the antibiotics and therefore to higher resistance. They also show that sub-inhibitory concentrations of streptomycin reduce the natural transformation capacity of *H. pylori*, a phenotype partially reverted by inactivation of hopD or hopE. While some of the observations are interesting and the experiments in most cases are done rigorously, the manuscript has several issues that need to be addressed:

Major:

1) The writing is confusing and it is hard to draw the conclusions from the experiments. Furthermore, the text requires extensive English editing.

- 2) Fig 1A: While the text says that 7 antibiotics were tested, the streptomycin plates are not presented for the E-test. More importantly, only the hopD and hopE mutants show inhibition zones on the E-test while neither the wt nor the other porin mutants are inhibited. Unless I am missing something, there seems to be a contradiction with the reported MICs that are higher for the hopD and hopE mutants.
- 3) The design of the transformation experiments described in Fig C-F is inappropriate for addressing the question the authors ask about the inhibition of translation by streptomycin of the transforming DNA-coded RNA. The expression of the rpsL-GFP fusion requires the integration of its coding sequence into the bacterial genome, therefore, the lower GFP levels could also be due to a deficiency in any of the natural transformation steps, from the DNA capture to its recombination into the chromosome. Furthermore, their hypothesis that streptomycin sub-inhibitory concentrations reduce the translation of the ribosomal protein isn't supported by the RPLC-MS experiment on the wild type strain RpsL is not found amongst the down-regulated proteins.
- 4) While the authors get the answer to the question for general translation inhibition by streptomycin through the RPLC-MS experiment, if they wanted to test whether streptomycin is inhibiting translation using GFP expression as a parameter, they should integrate into the genome the GFP-coding sequence under the control of a constitutive promoter and monitor fluorescence in wt and hop mutants.
- 5) The discussion is extremely confusing and the conclusions are not clear. In particular, the proposed link between sub-inhibitory antibiotic exposure and transformation efficiency is not well explained. Indeed, the fact that the hopE mutant was able to partially compensate the effect on transformation of rifampicin (transcription inhibitor), and metronidazole (DNA damaging agent) but neither of clarithromycin or tetracyclin (both translation inhibitors) makes it unlikely that protein translation is the common mechanism.
- 6) The discussion should better summarize the main conclusions and should better distinguish what is known about *H. pylori* and other species. In particular, many of the results cited on the effect of antibiotics on HGT are from species where competence is known to be finely regulated, which, so far, doesn't seem to be the case for *H. pylori*.

Minor:

- 1) Fig 2: There are discrepancies regarding the incubation time in the presence of the fluorescent streptomycin between the description of the experiment in the Materials and Methods and that in the legend of the figure (4h and 1h, respectively). I could not find the black dots and triangles mentioned in the legend.
- 2) lines 268-269: as the authors mention in other parts of the text, (transcription inhibitor) and metronidazole (DNA damaging agent) are not protein synthesis inhibitors.
- 3) Some of the references do not correspond to what the authors say (i.e. reference 21 (Bina et al., 2000) should be replaced by (M. M. Exner, P. Doig, T. J. Trust, and R. E. W. Hancock, *Infect. Immun.* 63:1567-1572, 1995) ; in references 10, 17 and 18 the transmission of streptomycin-resistance genes between strains is not analysed.

In conclusion, this manuscript demonstrates that HopE, and to a minor extent HopD porins, contribute to the susceptibility to sub-inhibitory concentrations of antibiotics by facilitating the influx, at least in the case of streptomycin. This could be better expressed in the title. The mechanisms underlying the effect of antibiotics on natural transformation are however less clear and are not well documented.

Reviewer #2 (Comments for the Author):

In the manuscript entitled "The effect of porins in the intrinsic antimicrobial tolerance and resistance acquisition by natural transformation in *Helicobacter pylori*" the authors have studied the significance of *H. pylori* porins in the context of antimicrobial resistance. Although their findings on the role of HopD and HopE in relation to antimicrobial resistance are interesting several important controls are necessary in order to prove the drawn conclusions.

Major comments:

1. Porins are known important molecules in the context of β -lactam and fluoroquinolone antibiotics. In the Figure 1A-W, an antibiotic should also be used that is not affected by HopD and HopE mutations. This is required in order to show specificity.
2. Page 8, line 169 and Figure 2: "HopD and HopE perform influx of a certain amount of streptomycin to affect the uptake of the antibiotic in *H. pylori*". Only, wild type, hopD and hopE mutants were used. Using other hop mutants are necessary to check if there is any effect.
3. Figure 3 and Figure 4 can be merged since that will give the reader a better view of the effect of hopD and hopE mutations as compared to wild type in the presence or in the absence of streptomycin. Since the other hop mutants are available, the authors should use them in order to show specificity. It is worth mentioning that the authors have said "Interestingly, we found that, for the isogenic StrepR DNA transformation, the mutant recipient cells were still able to acquire a degree of resistance in the presence of streptomycin..." in page 11, line 227.
4. Is the difference that was obtained in the Figure 5 significant?
5. The authors are requested to provide a table showing the MIC of the strain 26695 wild type and the mutants. A different table should show the suboptimal concentrations that were used in the Figure 11-W.

Minor comments:

1. Please mention *Helicobacter pylori* when it is mentioned first time in the abstract, importance and in introduction. For the other parts *H. pylori* is fine.
2. Page 2, line 25-26: "Antibiotic resistance of both intrinsic and acquired origins is the ability of *H. pylori* to survive and thrive under antibiotics." Please rephrase the sentence with more clarity.
3. Page 2, line 35: Please give the expansion of TRITC.
4. Page 3, line 45: It should be the genes, *hopE* and *hopD*.
5. Page 4, line 86: Please give the full form of AR.
6. Page 5, line 110: Please remove the space between "Gram- negative" and write it as "Gram-negative".
7. Page 6, line 111: Remove the space between "membrane- associated" and write it as "membrane-associated".
8. Page 8, line 172: "be" is missing after "may".
9. Page 9, line 184: Gene *rpsL* should be in italics throughout the manuscript.
10. Page 10, line 201: change "reduced" to "reduction of".
11. Page 13, line 278: The sentence can be modified to 'Antibiotic uptake and efflux by *H. pylori* are basis for intrinsic susceptibility to antibiotics.'
12. Page 14, line 299: change "hydrophobic of" to either "hydrophobic property of" or "hydrophobic nature of".
13. Page 15, line 321: change "et al" to "et al."
14. Page 15, line 321: change "reduced nearly" to "reduced to nearly".
15. Page 16, line 347: Rifampin inhibits transcription.
16. Page 18, line 375: The microaerobic condition for *H. pylori* is usually 5% oxygen, 10% carbon dioxide and 85% nitrogen. The incubation temperature is 37{degree sign}C. In few areas the temperature used is mentioned as 35{degree sign}C.
17. Page 18, line 378: Insert comma after blood.
18. Page 18, line 386 and Page 18, line 390: 'Antibiotics susceptibility' is better than 'antimicrobial resistance'.
19. Page 18, line 389: Use "by agar dilution method" instead of "an agar dilution method".
20. Page 19, line 415: Use "appropriate for *hopA*" instead of "appropriate of *hopA*".
21. Page 20, line 438: Write *E. coli* in italics.
22. Page 21, line 443-444 and 452-453: Is it Defibrillated FBS or Defibrinated blood? Serum does not need defibrination. If it is blood, how come it is decomplexed?
23. Page 23, line 489: Use square bracket as the outer one.
24. Page 24, line 511: Provide the concentrations of containing phosphatase inhibitor, protease inhibitor and MSF. Also please provide the expansion of MSF.
25. Page 24, line 512: "...were subjected to 8 times sonication (50W power) and protein dissolution buffer". It should be "in protein dissociation buffer".
26. Figure numbers are not provided in along with the figures.
27. Please remove the space between magnitude of temperature and {degree sign}C seen in different places throughout the manuscript.

Staff Comments:

Preparing Revision Guidelines

Please return the manuscript within 60 days; if you cannot complete the modification within this time period, please contact me. If you do not wish to modify the manuscript and prefer to submit it to another journal, please notify me of your decision

immediately so that the manuscript may be formally withdrawn from consideration by Microbiology Spectrum.

In the manuscript entitled “The effect of porins in the intrinsic antimicrobial tolerance and resistance acquisition by natural transformation in *Helicobacter pylori*” the authors have studied the significance of *H. pylori* porins in the context of antimicrobial resistance. Although their findings on the role of HopD and HopE in relation to antimicrobial resistance are interesting several important controls are necessary in order to prove the drawn conclusions.

Major comments:

1. Porins are known important molecules in the context of β -lactam and fluoroquinolone antibiotics. In the Figure 1A-W, an antibiotic should also be used that is not affected by HopD and HopE mutations. This is required in order to show specificity.
2. Page 8, line 169 and Figure 2: “HopD and HopE perform influx of a certain amount of streptomycin to affect the uptake of the antibiotic in *H. pylori*”. Only, wild type, hopD and hopE mutants were used. Using other hop mutants are necessary to check if there is any effect.
3. Figure 3 and Figure 4 can be merged since that will give the reader a better view of the effect of hopD and hopE mutations as compared to wild type in the presence or in the absence of streptomycin. Since the other hop mutants are available, the authors should use them in order to show specificity. It is worth mentioning that the authors have said “Interestingly, we found that, for the isogenic StrepR DNA transformation, the mutant recipient cells were still able to acquire a degree of resistance in the presence of streptomycin...” in page 11, line 227.
4. Is the difference that was obtained in the Figure 5 significant?
5. The authors are requested to provide a table showing the MIC of the strain 26695 wild type and the mutants. A different table should show the suboptimal concentrations that were used in the Figure 1I-W.

Minor comments:

1. Please mention *Helicobacter pylori* when it is mentioned first time in the abstract, importance and in introduction. For the other parts *H. pylori* is fine.
2. **Page 2, line 25-26:** “Antibiotic resistance of both intrinsic and acquired origins is the ability of *H. pylori* to survive and thrive under antibiotics.” Please rephrase the sentence with more clarity.
3. **Page 2, line 35:** Please give the expansion of TRITC.
4. **Page 3, line 45:** It should be the genes, *hopE* and *hopD*.
5. **Page 4, line 86:** Please give the full form of AR.

6. **Page 5, line 110:** Please remove the space between “Gram- negative” and write it as “Gram-negative”.
7. **Page 6, line 111:** Remove the space between “membrane- associated” and write it as “membrane-associated”.
8. **Page 8, line 172:** “be” is missing after “may”.
9. **Page 9, line 184:** Gene *rpsL* should be in italics throughout the manuscript.
10. **Page 10, line 201:** change “reduced” to “reduction of”.
11. **Page 13, line 278:** The sentence can be modified to ‘Antibiotic uptake and efflux by *H. pylori* are basis for intrinsic susceptibility to antibiotics.’
12. **Page 14, line 299:** change “hydrophobic of” to either “hydrophobic property of” or “hydrophobic nature of”.
13. **Page 15, line 321:** change “et al” to “et al.”.
14. **Page 15, line 321:** change “reduced nearly” to “reduced to nearly”.
15. **Page 16, line 347:** Rifampin inhibits transcription.
16. **Page 18, line 375:** The microaerobic condition for *H. pylori* is usually 5% oxygen, 10% carbon dioxide and 85% nitrogen. The incubation temperature is 37°C. In few areas the temperature used is mentioned as 35°C.
17. **Page 18, line 378:** Insert comma after blood.
18. **Page 18, line 386 and Page 18, line 390:** ‘Antibiotics susceptibility’ is better than ‘antimicrobial resistance’.
19. **Page 18, line 389:** Use “by agar dilution method” instead of “an agar dilution method”.
20. **Page 19, line 415:** Use “appropriate for *hopA*” instead of “appropriate of *hopA*”.
21. **Page 20, line 438:** Write *E. coli* in italics.
22. **Page 21, line 443-444 and 452-453:** Is it Defibrillated FBS or Defibrinated blood? Serum does not need defibrination. If it is blood, how come it is decomplemented?
23. **Page 23, line 489:** Use square bracket as the outer one.
24. **Page 24, line 511:** Provide the concentrations of containing phosphatase inhibitor, protease inhibitor and MSF. Also please provide the expansion of MSF.
25. **Page 24, line 512:** “...were subjected to 8 times sonication (50W power) and protein dissolution buffer”. It should be “in protein dissociation buffer”.
26. Figure numbers are not provided in along with the figures.
27. Please remove the space between magnitude of temperature and °C seen in different places throughout the manuscript.

Reviewer comments:

Reviewer #1 (Comments for the Author):

In this manuscript, Liu et al. explore the consequences of inactivating porins on the antibiotic susceptibility and the natural transformation capacity of *H. pylori*. For that, they generated mutant strains defective in each of five different porin genes (hopA-E) and determined their MIC for 7 different antibiotics. They also monitored the capacity of the mutants to uptake a fluorescent analogue of streptomycin. From these experiments the authors concluded that porins HopE and D absence leads to a reduced influx of the antibiotics and therefore to higher resistance. They also show that sub-inhibitory concentrations of streptomycin reduce the natural transformation capacity of *H. pylori*, a phenotype partially reverted by inactivation of hopD or hopE.

While some of the observations are interesting and the experiments in most cases are done rigorously, the manuscript has several issues that need to be addressed:

Major:

- 1) The writing is confusing and it is hard to draw the conclusions from the experiments. Furthermore, the text requires extensive English editing.
- 2) Fig 1A: While the text says that 7 antibiotics were tested, the streptomycin plates are not presented for the E-test. More importantly, only the hopD and hopE mutants show inhibition zones on the E-test while neither the wt nor the other porin mutants are inhibited. Unless I am missing something, there seems to be a contradiction with the reported MICs that are higher for the hopD and hopE mutants.
- 3) The design of the transformation experiments described in Fig C-F is inappropriate for addressing the question the authors ask about the inhibition of translation by streptomycin of the transforming DNA-coded RNA. The expression of the rpsL-GFP fusion requires the integration of its coding sequence into the bacterial genome, therefore, the lower GFP levels could also be due to a deficiency in any of the natural transformation steps, from the DNA capture to its recombination into the chromosome. Furthermore, their hypothesis that streptomycin sub-inhibitory concentrations reduce the translation of the ribosomal protein isn't supported by the RPLC-MS experiment on the wild type strain RpsL is not found amongst the down-regulated proteins.
- 4) While the authors get the answer to the question for general translation inhibition by streptomycin through the RPLC-MS experiment, if they wanted to test whether streptomycin is inhibiting translation using GFP expression as a parameter, they should integrate into the genome the GFP-coding sequence under the control of a constitutive promoter and monitor fluorescence in wt and hop mutants.

5) The discussion is extremely confusing and the conclusions are not clear. In particular, the proposed link between sub-inhibitory antibiotic exposure and transformation efficiency is not well explained. Indeed, the fact that the hopE mutant was able to partially compensate the effect on transformation of rifampicin (transcription inhibitor), and metronidazole (DNA damaging agent) but neither of clarithromycin or tetracyclin (both translation inhibitors) makes it unlikely that protein translation is the common mechanism.

6) The discussion should better summarize the main conclusions and should better distinguish what is known about *H. pylori* and other species. In particular, many of the results cited on the effect of antibiotics on HGT are from species where competence is known to be finely regulated, which, so far, doesn't seem to be the case for *H. pylori*.

Minor:

1) Fig 2: There are discrepancies regarding the incubation time in the presence of the fluorescent streptomycin between the description of the experiment in the Materials and Methods and that in the legend of the figure (4h and 1h, respectively). I could not find the black dots and triangles mentioned in the legend.

2) lines 268-269: as the authors mention in other parts of the text, (transcription inhibitor) and metronidazole (DNA damaging agent) are not protein synthesis inhibitors.

3) Some of the references do not correspond to what the authors say (i.e. reference 21 (Bina et al., 2000) should be replaced by (M. M. Exner, P. Doig, T. J. Trust, and R. E. W. Hancock, *Infect. Immun.* 63:1567-1572, 1995) ; in references 10, 17 and 18 the transmission of streptomycin-resistance genes between strains is not analysed.

In conclusion, this manuscript demonstrates that HopE, and to a minor extent HopD porins, contribute to the susceptibility to sub-inhibitory concentrations of antibiotics by facilitating the influx, at least in the case of streptomycin. This could be better expressed in the title. The mechanisms underlying the effect of antibiotics on natural transformation are however less clear and are not well documented.

Reviewer #2 (Comments for the Author):

In the manuscript entitled "The effect of porins in the intrinsic antimicrobial tolerance and resistance acquisition by natural transformation in *Helicobacter pylori*" the authors have studied the significance of *H. pylori* porins in the

context of antimicrobial resistance. Although their findings on the role of HopD and HopE in relation to antimicrobial resistance are interesting several important controls are necessary in order to prove the drawn conclusions.

Major comments:

1. Porins are known important molecules in the context of β -lactam and fluoroquinolone antibiotics. In the Figure 1A-W, an antibiotic should also be used that is not affected by HopD and HopE mutations. This is required in order to show specificity.
2. Page 8, line 169 and Figure 2: "HopD and HopE perform influx of a certain amount of streptomycin to affect the uptake of the antibiotic in *H. pylori*". Only, wild type, hopD and hopE mutants were used. Using other hop mutants are necessary to check if there is any effect.
3. Figure 3 and Figure 4 can be merged since that will give the reader a better view of the effect of hopD and hopE mutations as compared to wild type in the presence or in the absence of streptomycin. Since the other hop mutants are available, the authors should use them in order to show specificity. It is worth mentioning that the authors have said "Interestingly, we found that, for the isogenic StrepR DNA transformation, the mutant recipient cells were still able to acquire a degree of resistance in the presence of streptomycin..." in page 11, line 227.
4. Is the difference that was obtained in the Figure 5 significant?
5. The authors are requested to provide a table showing the MIC of the strain 26695 wild type and the mutants. A different table should show the suboptimal concentrations that were used in the Figure 1I-W.

Minor comments:

1. Please mention *Helicobacter pylori* when it is mentioned first time in the abstract, importance and in introduction. For the other parts *H. pylori* is fine.
2. Page 2, line 25-26: "Antibiotic resistance of both intrinsic and acquired origins is the ability of *H. pylori* to survive and thrive under antibiotics." Please rephrase the sentence with more clarity.
3. Page 2, line 35: Please give the expansion of TRITC.
4. Page 3, line 45: It should be the genes, hopE and hopD.
5. Page 4, line 86: Please give the full form of AR.
6. Page 5, line 110: Please remove the space between "Gram- negative" and write it as "Gram-negative".
7. Page 6, line 111: Remove the space between "membrane- associated" and write it as "membrane-associated".
8. Page 8, line 172: "be" is missing after "may".
9. Page 9, line 184: Gene *rpsL* should be in italics throughout the manuscript.
10. Page 10, line 201: change "reduced" to "reduction of".

11. Page 13, line 278: The sentence can be modified to 'Antibiotic uptake and efflux by *H. pylori* are basis for intrinsic susceptibility to antibiotics.'
12. Page 14, line 299: change "hydrophobic of" to either "hydrophobic property of" or "hydrophobic nature of".
13. Page 15, line 321: change "et al" to "et al."
14. Page 15, line 321: change "reduced nearly" to "reduced to nearly".
15. Page 16, line 347: Rifampin inhibits transcription.
16. Page 18, line 375: The microaerobic condition for *H. pylori* is usually 5% oxygen, 10% carbon dioxide and 85% nitrogen. The incubation temperature is 37{degree sign}C. In few areas the temperature used is mentioned as 35{degree sign}C.
17. Page 18, line 378: Insert comma after blood.
18. Page 18, line 386 and Page 18, line 390: 'Antibiotics susceptibility' is better than 'antimicrobial resistance'.
19. Page 18, line 389: Use "by agar dilution method" instead of "an agar dilution method".
20. Page 19, line 415: Use "appropriate for hopA" instead of "appropriate of hopA".
21. Page 20, line 438: Write *E. coli* in italics.
22. Page 21, line 443-444 and 452-453: Is it Defibrillated FBS or Defibrinated blood? Serum does not need defibrination. If it is blood, how come it is de complemented?
23. Page 23, line 489: Use square bracket as the outer one.
24. Page 24, line 511: Provide the concentrations of containing phosphatase inhibitor, protease inhibitor and MSF. Also please provide the expansion of MSF.
25. Page 24, line 512: "...were subjected to 8 times sonication (50W power) and protein dissolution buffer". It should be "in protein dissociation buffer".
26. Figure numbers are not provided in along with the figures.
27. Please remove the space between magnitude of temperature and {degree sign}C seen in different places throughout the manuscript.

Point by point answer to comments

Major changes in the text are indicated in red.

Reviewer 1

In this manuscript, Liu et al. explore the consequences of inactivating porins on the antibiotic susceptibility and the natural transformation capacity of H. pylori. For that, they generated mutant strains defective in each of five different porin genes (hopA-E) and determined their MIC for 7 different antibiotics. They also monitored the capacity of the mutants to uptake a fluorescent analogue of streptomycin. From these experiments the authors concluded that porins HopE and D absence leads to a reduced influx of the antibiotics and therefore to higher resistance. They also show that sub-inhibitory concentrations of streptomycin reduce the natural transformation capacity of H. pylori, a phenotype partially reverted by inactivation of hopD or hopE. While some of the observations are interesting and the experiments in most cases are done rigorously, the manuscript has several issues that need to be addressed:

We thank this reviewer for the positive evaluation of our results.
We have significantly improved the manuscript following the reviewer's advices.

Major:

1) The writing is confusing and it is hard to draw the conclusions from the experiments. Furthermore, the text requires extensive English editing.

Thanks very much for your comments. We have carefully checked and modified the paper to more accurately describe the meaning and intention of the experimental results. And we have improved and polished the English writing in our revised manuscript. We hope the revised version meet the English presentation standard.

2) Fig 1A: While the text says that 7 antibiotics were tested, the streptomycin plates are not presented for the E-test.

In this study, the antimicrobial susceptibility of *H. pylori* 26695 and mutants to six antibiotics (metronidazole, clarithromycin, levofloxacin, amoxicillin, tetracycline and rifampin) were determined by both E-test (Figure 1A) and an agar dilution method (Figure 1B-F, H), and the antimicrobial susceptibility to streptomycin was determined by an agar dilution method only (Figure 1G). This was demonstrated already in the Determination of antimicrobial susceptibility part (new page 24, new line 502-508) of the Materials and methods section. There are two reasons explaining the lack of the E-test method in determining

the susceptibility to streptomycin: 1) there is no commercial streptomycin E-test strip unavailable; 2) the agar dilution method is considered the most traditional approach to test the antimicrobial susceptibility in *H. pylori*, and it is widely accepted that this single method is able to determine the MIC value of *H. pylori* (Hulten et al, Gastroenterology, 2021; Lee et al, Gut Liver, 2021; Seo et al, BMC Gastroenterol, 2019).

More importantly, only the hopD and hopE mutants show inhibition zones on the E-test while neither the wt nor the other porin mutants are inhibited. Unless I am missing something, there seems to be a contradiction with the reported MICs that are higher for the hopD and hopE mutants.

For the second question, there were no visible inhibition zones on the E-test plates tested for WT and the other porin mutants. Because the drug concentrations on the test strips and the MH plates decrease from upper to below, this result means that the strains were very sensitive to the tested antibiotics, and even the lowest concentration on the bottom of the plates was able to inhibit the bacteria growth, in other word, the MIC values were too low to allow these strains to grow on the plates. These results were consistent with the reports of higher MICs for the hopD and hopE mutants.

3) The design of the transformation experiments described in Fig C-F is inappropriate for addressing the question the authors ask about the inhibition of translation by streptomycin of the transforming DNA-coded RNA. The expression of the rpsL-GFP fusion requires the integration of its coding sequence into the bacterial genome, therefore, the lower GFP levels could also be due to a deficiency in any of the natural transformation steps, from the DNA capture to its recombination into the chromosome.

We agree with the reviewer on the idea that the expression of the transforming gene fusion could be affected by any of the natural transformation steps, from the DNA capture to its recombination into the chromosome. However, this experiment controlled a single variable in the two groups, that is, adding or not adding streptomycin, and kept other factors the same, including the concentration of extracellular rpsL-GFP fusion, incubation time, and incubation condition. Therefore, we think that the differential GFP expression was caused by the single variable (the influence of streptomycin), and other factors in the process of natural transformation and homologous recombination which were kept consistent in two groups are beyond the scope of this work. This design of the transformation experiments is supported by several studies investigating the effect of drugs on the expression of acquired gene such as the study by Nolivos et al (Nolivos et al, Science, 2019). Secondly, the *H. pylori* chromosome genome per se does not contain the GFP gene, so the detected green fluorescent GFP is considered the product of the successful uptake, integration

and expression of the foreign gene. Thus we added this negative control in the Revised Figure 3E and corresponding content in the Result section (new page 10, new line 214-218) and Figure legend.

Furthermore, their hypothesis that streptomycin sub-inhibitory concentrations reduce the translation of the ribosomal protein isn't supported by the RPLC-MS experiment on the wild type strain RpsL is not found amongst the down-regulated proteins.

Thank you for your valuable comment. In our RPLC-MS analysis result, rpsL is indeed down-regulated but did not reach a significant decrease in abundance. This disparity could be likely explained by the primary limitations in label-free proteomics that it has difficulties in ensuring the quantification reproducibility and robustness, and has obstacles to achieving its best sensitivity (Zhang et al, J Proteome Res, 2020; Klaene et al, J Chromatogr A, 2016). Similar disparity was also manifested in the aspect that a total of 878 and 859 proteins were detected in WT strain and $\Delta hopE$ mutant cells, respectively, while there are about 1500 coding genes on average in *H. pylori* chromosome genomes, such as 1590 coding genes in *H. pylori* 26695. Moreover, compared with the high-throughput proteomics analysis technology, quantification of the expression specific to this protein seems more confirming. We have added the relevant discussion in the Discussion chapter (new page 19-20, new line 414-420).

4) While the authors get the answer to the question for general translation inhibition by streptomycin through the RPLC-MS experiment, if they wanted to test whether streptomycin is inhibiting translation using GFP expression as a parameter, they should integrate into the genome the GFP-coding sequence under the control of a constitutive promoter and monitor fluorescence in wt and hop mutants.

In this study, the rpsLmut-GFP fusion (*gfp* in frame with *rpsL*) was constructed to monitor rpsLmut expression. Moreover, as the reviewer suggested, we added the GFP expression as a parameter to test whether streptomycin was inhibiting translation. We transformed the GFP-coding sequence into the *H. pylori* under HpUreA promoter and monitored the fluorescence in WT and $\Delta hopE$ mutant. The result was added in the revised Supplementary Material as Fig. S3 and in the Result section (new page 12, new line 257-263).

Supplementary Figure S3

5) The discussion is extremely confusing and the conclusions are not clear. In particular, the proposed link between sub-inhibitory antibiotic exposure and transformation efficiency is not well explained.

We have reorganized the discussion section according to the results. The conclusion was rewritten and extracted as the Conclusion section at the end of Discussion (new page 21-22, new line 453 to 463). Moreover, the reason explaining the proposed inhibiting effect of sub-inhibitory streptomycin exposure on transformation efficiency in our second part result (Figure 2) has been mentioned in the appropriate place (new page 18, new line 380 to 382) and specifically discussed in terms our results in the next paragraph.

Indeed, the fact that the hopE mutant was able to partially compensate the effect on transformation of rifampicin (transcription inhibitor), and metronidazole (DNA damaging agent) but neither of clarithromycin or tetracyclin (both translation inhibitors) makes it unlikely that protein translation is the common mechanism.

Thank you for your thoughtful comment, and we improved the discussion according to your question.

From our study, the acquired gene expression observed after transformation bound to be affected by the intracellular accumulation of the translation- or transcription-inhibiting antibiotics such as streptomycin and rifampicin. Therefore, in the presence of these antibiotics, they could be the sufficient drug accumulation in WT strain and reduced drug concentration in mutant that

caused the significant difference in NT frequency (and protein expression). While in the presence of clarithromycin or tetracycline, the unobserved mechanism that *hopE* deletion did not significantly restore NT frequency was actually due to the same high level of NT frequency in WT strain, which could be due to the insufficient drug accumulation in WT strain. It is known that the intracellular levels of antibiotics are regulated by the balance of two main factors: porin-mediated influx and efflux pump mediated efflux, which have their own substrates. Therefore, it could be speculated that the sufficient accumulation of streptomycin and rifampicin in WT strain may be due to the lack of efficiently efflux of these antibiotics, and the insufficient accumulation of clarithromycin or tetracycline in WT strain was probably due to the efficient efflux of the two antibiotics by efflux pump in *H. pylori*. Perhaps, clarithromycin and tetracycline rather than rifampicin and metronidazole are within the substrates of the main efflux pump in *H. pylori*.

We have added this discussion in the appropriate place (new page 20-21, new line 434 to 451).

6) The discussion should better summarize the main conclusions and should better distinguish what is known about H. pylori and other species. In particular, many of the results cited on the effect of antibiotics on HGT are from species where competence is known to be finely regulated, which, so far, doesn't seem to be the case for H. pylori.

As the reviewer suggested, the conclusion has been summarized at the end of the Discussion section (new page 21-22, new line 453 to 463). Currently relevant studies on the role of antibiotics on HGT induced resistance in different species have been summarized and compared with what is known in *H. pylori* (new page 15-17, new line 328 to 371). Moreover, we have explained that, instead of concentrating on the transformation efficiency affected by DNA-damaging antibiotic utilization investigated in current other studies, which were not within the scope of this study, we chose another perspective, focusing on the effect of cellular accumulation of antibiotics, especially protein synthesis-inhibiting antibiotics, controlled by putative porins, on resistance acquisition via directly detecting the NT frequency and acquired resistance gene expression (new page 16-17, new line 372 to 378).

Minor:

1) Fig 2: There are discrepancies regarding the incubation time in the presence of the fluorescent streptomycin between the description of the experiment in the Materials and Methods and that in the legend of the figure (4h and 1h, respectively). I could not find the black dots and triangles mentioned in the legend.

Sorry for the wrong incubation time in the Materials and Methods section and the lost triangle in Figure 2B by mistake. The corrected 1 h incubation time has been modified (new page 28, new line 589), and the triangles have also been added in the revised Figure 2B.

2) lines 268-269: as the authors mention in other parts of the text, (transcription inhibitor) and metronidazole (DNA damaging agent) are not protein synthesis inhibitors.

Rifampin, clarithromycin and tetracycline can inhibit protein synthesis at transcription and translation levels, respectively. Here we wanted to state that “three protein synthesis-inhibiting antibiotics clarithromycin, tetracycline and rifampin, and a DNA damaging agent metronidazole”. We have rewritten this sentence and made it clear (new page 13, new line 283).

3) Some of the references do not correspond to what the authors say (i.e. reference 21 (Bina et al., 2000) should be replaced by (M. M. Exner, P. Doig, T. J. Trust, and R. E. W. Hancock, Infect. Immun. 63:1567-1572, 1995) ; in references 10, 17 and 18 the transmission of streptomycin-resistance genes between strains is not analysed.

Thank you so much for your careful check. The reference 21 (Bina et al., 2000) has been replaced by (M. M. Exner, P. Doig, T. J. Trust, and R. E. W. Hancock, Infect. Immun. 63:1567-1572, 1995).

In the reference 10, the transformation efficiency of bacteria grown in liquid with PCR fragments conferring streptomycin resistance was measured. In the reference 17, a DNA sequence coding for streptomycin resistance was used to determine the NT frequency. In the reference 18, genomic DNA conferring streptomycin resistance was used in the transformation test. Therefore, we modified the sentence “NT contributes to the transmission of streptomycin-resistance associated genes among *H. pylori* strains” to “NT enables *H. pylori* strains to transmit streptomycin resistance-conferring DNA fragment” (new page 5, new line 100-101).

In conclusion, this manuscript demonstrates that HopE, and to a minor extent HopD porins, contribute to the susceptibility to sub-inhibitory concentrations of antibiotics by facilitating the influx, at least in the case of streptomycin. This could be better expressed in the title.

As the reviewer suggested, the title has been modified to “HopE and HopD porins-mediated drug influx contribute to the intrinsic antimicrobial susceptibility and inhibit streptomycin resistance acquisition by natural transformation in *Helicobacter pylori*”.

The mechanisms underlying the effect of antibiotics on natural transformation are however less clear and are not well documented.

We agree with this reviewer on the need to further explore the mechanisms underlying the effect of antibiotics on the process of natural transformation per se from DNA uptake to integration, but it is not within the scope of our study.

In this study, the mechanism underlying the effect of antibiotics on the antibiotic resistance acquisition by NT was investigated from the perspective of the role of porin-mediated influx for the first time. We hope the discussion below can well summary the main findings of our study.

“Our study preliminarily revealed the important role of HopE, and to a minor extent HopD porins, in evading acquired antibiotic resistance by NT in the presence of translation-inhibiting antimicrobial streptomycin. In the presence of streptomycin, resistance acquisition at least partially relied on the loss of HopE and HopD, because their deletions decreased drug concentration in the cell, and thus restored the expression of the resistance-conferring gene which was inhibited by streptomycin in wild-type strain. Therefore, the absence of HopE and/or HopD porins in *H. pylori* genomes, combined with the large number of secreted or cell-free genetic elements carrying mutations conferring antibiotic resistance, may raise the possibility that this mechanism plays a potential role in the dissemination of drug resistance within *H. pylori* communities”.

Once again, we thank this reviewer for his/her comments.

Reviewer 2:

In the manuscript entitled "The effect of porins in the intrinsic antimicrobial tolerance and resistance acquisition by natural transformation in Helicobacter pylori" the authors have studied the significance of H. pylori porins in the context of antimicrobial resistance. Although their findings on the role of HopD and HopE in relation to antimicrobial resistance are interesting several important controls are necessary in order to prove the drawn conclusions.

We thank this reviewer for the positive evaluation of our results.

We have significantly improved the manuscript following the reviewer's advices.

Major comments:

1. Porins are known important molecules in the context of β -lactam and fluoroquinolone antibiotics. In the Figure 1A-W, an antibiotic should also be used that is not affected by HopD and HopE mutations. This is required in order to show specificity.

In our antimicrobial susceptibility determination by E-test method, chloramphenicol was not affected by hopD and hopE mutations. To show specificity, we have added the result in the revised Figure 1A and the corresponding content in the manuscript.

2. Page 8, line 169 and Figure 2: "HopD and HopE perform influx of a certain amount of streptomycin to affect the uptake of the antibiotic in *H. pylori*". Only, wild type, hopD and hopE mutants were used. Using other hop mutants are necessary to check if there is any effect.

We have previously performed the same experiments in $\Delta hopA$, $\Delta hopB$ and $\Delta hopC$ mutants. Given that the result showed no involvement of HopA, HopB and HopC in streptomycin influx, we did not put them in the manuscript. Here as suggested by the reviewer, the representatives of the red fluorescence detection after 1 h incubation in the presence of TRITC labeled-streptomycin and quantification comparisons of streptomycin uptake by WT, $\Delta hopA$, $\Delta hopB$ and $\Delta hopC$ strains were added in the revised Supplementary Material as Fig. S1 and in the Result section (new page 8, new line 174-176).

Supplementary Figure S1

3. Figure 3 and Figure 4 can be merged since that will give the reader a better view of the effect of hopD and hopE mutations as compared to wild type in the presence or in the absence of streptomycin. Since the other hop mutants are available, the authors should use them in order to show specificity. It is worth mentioning that the authors have said "Interestingly, we found that, for the isogenic StrepR DNA transformation, the mutant recipient cells were still able to

acquire a degree of resistance in the presence of streptomycin..." in page 11, line 227.

We have merged Figure 3 and Figure 4 as a revised Figure 3 in which the comparisons of transformation frequency (revised figure 3A, B) and the rpsLmut-GFP intracellular fluorescence (revised figure 3G) between WT strain and mutants were clearly showed in one graph. The corresponding content was also modified in the text. By merging the two figures, the reviewer mentioned "*Interestingly, we found that, for the isogenic StrepR DNA transformation, the mutant recipient cells were still able to acquire a degree of resistance in the presence of streptomycin..."*" can be showed more intuitively.

To show specificity, other three porins-deleted mutants were also used to measure the transformation frequency, which showed that, in the presence of streptomycin, $\Delta hopA$, $\Delta hopB$ and $\Delta hopC$ mutants all showed unchanged NT frequencies compared with that in WT strain when incubated with neither isogenic StrepR DNA nor pUC19-*rpsLmut-sfgfp* linear plasmid. The result has been added in the revised Supplementary Material as Fig. S2 and in the Result section (new page 12, new line 244-248).

Supplementary Figure S2

4. Is the difference that was obtained in the Figure 5 significant?

Yes, the difference of the percentage of downregulated proteins between WT and $\Delta hopE$ strains was significantly ($p < 0.0001$) by statistical analysis using Chi-square test. The statistical analysis result has been added in the Revised figure 4 and corresponding content in the Result section (new page 13, new line 276) and the figure legend.

5. The authors are requested to provide a table showing the MIC of the strain 26695 wild type and the mutants. A different table should show the suboptimal concentrations that were used in the Figure 1I-W.

The source data of the MIC values were provided as a table in the Source Data file for Figure 1B-H. The suboptimal concentrations used in the Figure 1I-W were

provided as Supplementary Table S1 in the Supplementary Material file.

Minor comments:

1. Please mention Helicobacter pylori when it is mentioned first time in the abstract, importance and in introduction. For the other parts H. pylori is fine.

The full name *Helicobacter pylori* has now been mentioned when it firstly occurred in the abstract, importance and in introduction (new line 23, 50 and 87).

2. Page 2, line 25-26: "Antibiotic resistance of both intrinsic and acquired origins is the ability of H. pylori to survive and thrive under antibiotics." Please rephrase the sentence with more clarity.

The sentence has been rephrased as "Intrinsic and acquired antibiotic resistance contribute to the survival and multiplication of *H. pylori* under antibiotics (new line 24-25)".

3. Page 2, line 35: Please give the expansion of TRITC.

The expansion of TRITC has been given in the abstract (new line 35).

4. Page 3, line 45: It should be the genes, hopE and hopD.

HopE and HopD have been corrected to *hopE* and *hopD* genes (new line 44).

5. Page 4, line 86: Please give the full form of AR.

The full form of AR "antibiotic resistance" has been given (new line 86).

6. Page 5, line 110: Please remove the space between "Gram- negative" and write it as "Gram-negative".

Thank you very much for your careful check, the space between "Gram-negative" has been removed, and the phrase has been "Gram-negative" (new line 109).

7. Page 6, line 111: Remove the space between "membrane- associated" and write it as "membrane-associated".

The space between "membrane- associated" has been removed, and the phrase has been "membrane-associated" (new line 110).

8. Page 8, line 172: "be" is missing after "may".

The "be" after "may" has been added (new line 171).

9. Page 9, line 184: Gene *rpsL* should be in italics throughout the manuscript.

The gene *rpsL* has been changed to italics form when it represented a gene throughout the manuscript.

10. Page 10, line 201: change "reduced" to "reduction of".

The "reduced" has been changed to "reduction of" (new line 203).

11. Page 13, line 278: The sentence can be modified to 'Antibiotic uptake and efflux by *H. pylori* are basis for intrinsic susceptibility to antibiotics.'

The sentence has been modified to 'Antibiotic uptake and efflux by *H. pylori* are basis for intrinsic susceptibility to antibiotics' (new line 292).

12. Page 14, line 299: change "hydrophobic of" to either "hydrophobic property of" or "hydrophobic nature of".

We have changed the "hydrophobic of" to "hydrophobic property of" (new line 314).

13. Page 15, line 321: change "et al" to "et al.".

We have changed "et al" to "et al." (new line 388).

14. Page 15, line 321: change "reduced nearly" to "reduced to nearly".

We have changed "reduced nearly" to "reduced to nearly" (new line 388).

15. Page 16, line 347: Rifampin inhibits transcription.

We have changed "In the presence of another translation-inhibiting antibiotic rifampin" to "In the presence of the transcription-inhibiting antibiotic rifampin" (new line 421).

16. Page 18, line 375: The microaerobic condition for *H. pylori* is usually 5% oxygen, 10% carbon dioxide and 85% nitrogen. The incubation temperature is 37{degree sign}C. In few areas the temperature used is mentioned as 35{degree sign}C.

Sorry for the wrong descriptions. The "10% O₂, 5% CO₂, and 85% N₂" has been changed to "5% O₂, 10% CO₂, and 85% N₂", and "35 °C" has been changed to "37 °C" (new line 467 and 469).

17. Page 18, line 378: Insert comma after blood.

We have inserted a comma after "blood" (new line 470).

18. Page 18, line 386 and Page 18, line 390: 'Antibiotics susceptibility' is better than 'antimicrobial resistance'.

We have changed 'antimicrobial resistance' to 'antibiotics susceptibility' (new line 502 and 507).

19. Page 18, line 389: Use "by agar dilution method" instead of "an agar dilution method".

We have changed "an agar dilution method" to "by agar dilution method" (new line 505).

20. Page 19, line 415: Use "appropriate for hopA" instead of "appropriate of hopA".

We have changed "appropriate of hopA" to "appropriate for hopA" (new line 483).

21. Page 20, line 438: Write E. coli in italics.

We have written *E. coli* in italics (new line 531).

22. Page 21, line 443-444 and 452-453: Is it Defibrillated FBS or Defibrinated blood? Seru does not need defibrination. If it is blood, how come it is decomplemented?

Sorry for the wrong typing. It should be "defibrinated horse blood" and "decomplemented FBS". We have corrected all the wrong expressions throughout the manuscript. Specifically, the defibrinated horse blood was commercially obtained. For the decomplemented FBS, we got it just by incubating the serum in a water bath at 56 °C for 30 minutes. This is commonly used for the preparation of *H. pylori* liquid culture medium (Damke et al, nature communications, 2019).

23. Page 23, line 489: Use square bracket as the outer one.

We have replaced the outer parentheses with square bracket (new line 579-580).

24. Page 24, line 511: Provide the concentrations of containing phosphatase inhibitor, protease inhibitor and MSF. Also please provide the expansion of MSF.

The concentrations of phosphatase inhibitor, protease inhibitor and PMSF, and the expansion of PMSF have been added (new line 601-602). Actually, MSF should be PMSF, and it is a protease inhibitor.

25. Page 24, line 512: "...were subjected to 8 times sonication (50W power) and protein dissolution buffer". It should be "in protein dissociation buffer".

We have replaced "...were subjected to 8 times sonication (50W power) and protein dissolution buffer" by "...were subjected to 8 times sonication (50W power) and in protein dissolution buffer" (new line 603).

26. Figure numbers are not provided in along with the figures.

The Figure numbers were provided as the file name for each figure.

27. Please remove the space between magnitude of temperature and {degree sign}C seen in different places throughout the manuscript.

All the spaces between magnitude of temperature and °C throughout the manuscript have been removed.

December 22, 2021

Prof. Yanmei Zhang
Huadong Hospital affiliated to Fudan University
Department of Laboratory medicine
Shanghai
China

Re: Spectrum01987-21R1 (HopE and HopD porins-mediated drug influx contribute to the intrinsic antimicrobial susceptibility and inhibit streptomycin resistance acquisition by natural transformation in *Helicobacter pylori*)

Dear Prof. Yanmei Zhang:

Link Not Available

Sincerely,

Beile Gao

Journals Department
Reviewer comments:

Reviewer #1 (Comments for the Author):

- 1) Regarding comment 2) from the initial review, the data issued from Figure 1A is still confusing. The authors answered that for the wild type, hopA, hopB and hopC mutants there was no growth on the E-test plates. Not only that is not clear from the pictures (the non-growth zones look much darker for the hopD and E strains) but the text on lines 134-135 says that in those plates there is "no inhibition zone observed". I presume now it is a language problem. It needs to be clarified.
- 2) Labelling as sub-inhibitory the concentrations of the antibiotics in the experiments described in Figures 1I to 1V is confusing. In all cases the concentrations used completely inhibit growth of the wild-type strain.
- 3) Concerning point 3) response, while I agree that other factors are the same, what I meant is that streptomycin could affect natural transformation by, for example, changing the level of expression of some protein required for uptake or recombination of the transforming DNA. The results presented do not allow to rule out this kind of explanation.

Other comments:

While I appreciate that the authors have made a big effort to improve the text, there are still several English mistakes and some sentences make it very hard to understand the message and, in some cases, lead to misinterpretation.

The use of "certain amount" or "certain degree" throughout the text is inappropriate. What do the authors mean by that? A more rigorous language should be used.

Reviewer #2 (Comments for the Author):

None

Staff Comments:

Preparing Revision Guidelines

Please return the manuscript within 60 days; if you cannot complete the modification within this time period, please contact me. If you do not wish to modify the manuscript and prefer to submit it to another journal, please notify me of your decision immediately so that the manuscript may be formally withdrawn from consideration by Microbiology Spectrum.

Reviewer comments:

Reviewer #1 (Comments for the Author):

1) Regarding comment 2) from the initial review, the data issued from Figure 1A is still confusing. The authors answered that for the wild type, hopA, hopB and hopC mutants there was no growth on the E-test plates. Not only that is not clear from the pictures (the non-growth zones look much darker for the hopD and E strains) but the text on lines 134-135 says that in those plates there is "no inhibition zone observed". I presume now it is a language problem. It needs to be clarified.

To provide the clear image, the raw picture in original size has now been added in the Source Data file as a sheet for Figure 1A.

Non-growth areas look darker because the colony appearance is transparent/translucent, and the densely growing colonies formed a white film-like morphology on the plate, which made the growth zones look lighter.

Thank you very much for pointing out the inappropriate description. Now we have modified "no inhibition zone observed" to "no colony growth observed" in the Result section (new line 128).

2) Labelling as sub-inhibitory the concentrations of the antibiotics in the experiments described in Figures 1I to 1V is confusing. In all cases the concentrations used completely inhibit growth of the wild-type strain.

Sorry for the inconsistent labels. To avoid ambiguity, we have deleted the "sub-inhibitory concentrations" or revised it to "indicated concentrations" in the appropriate places in the manuscript (the modification traces were preserved in the Marked-Up Manuscript).

3) Concerning point 3) response, while I agree that other factors are the same, what I meant is that streptomycin could affect natural transformation by, for example, changing the level of expression of some protein required for uptake or recombination of the transforming DNA. The results presented do not allow to rule out this kind of explanation.

We are sorry for not understanding your meaning. We agree with the reviewer on the opinion that whether streptomycin could affect expression of some protein required for uptake or recombination of the transforming DNA is unknown. However, it seems unaffected or very slightly affected in our experiment here because: **1)** It seems not time-allowed. During the natural transformation incubation of strains with transforming DNA, streptomycin was

added at the same time with the transformation mix. While the whole process of transformation, from DNA capture to its expression after integration by recombination into the chromosome occurs in less than 1 h, and the DNA is fully expressed in 2-3 h (from this study and [1]), which relied on the existing levels of NT machinery proteins regardless of whether streptomycin affect these proteins' expression since they also need time to respond and alter their expressions or not. **2)** This concern may happen in this kind of experimental design: before strains mixing with transforming DNA, strains incubated with streptomycin for a certain time which allow the protein that may be affected by streptomycin fully expressed, and then adding the transforming DNA for natural transformation assay. However, in our experiment, streptomycin was added at the same time with the transformation mix.

On the other hand, this experiment design was employed by referring to another article in which the antibiotic was added at the same time than the conjugation mix to investigate the effect of the antibiotic on the expression of conjugated DNA from outside the recipient strains [2].

Now we have added the relevant discussion in the Discussion section (new line 409-417)

[1] Corbinais C, Mathieu A, Kortulewski T, Radicella JP, Marsin S. Following transforming DNA in *Helicobacter pylori* from uptake to expression. *Mol Microbiol.* 2016;101(6):1039-53.

[2] Nolivos S, Cayron J, Dedieu A, Page A, Delolme F, Lesterlin C. Role of AcrAB-TolC multidrug efflux pump in drug-resistance acquisition by plasmid transfer. *Science.* 2019;364(6442):778-782.

Other comments:

While I appreciate that the authors have made a big effort to improve the text, there are still several English mistakes and some sentences make it very hard to understand the message and, in some cases, lead to misinterpretation.

The use of "certain amount" or "certain degree" throughout the text is inappropriate. What do the authors mean by that? A more rigorous language should be used.

Sorry for the confusing statements. We meant to say "the reduced amount" and "the inhibited NT frequency", which caused semantic repetition and misinterpretation. Therefore, the "certain amount" and the "certain degree" has been deleted throughout the text (the modification traces were preserved in the Marked-Up Manuscript).

Once again, we thank this reviewer for his/her valuable comments.

Reviewer #2 (Comments for the Author):

None

We thank this reviewer for his/her recognition.

January 21, 2022

Prof. Yanmei Zhang
Huadong Hospital affiliated to Fudan University
Department of Laboratory medicine
Shanghai
China

Re: Spectrum01987-21R2 (HopE and HopD porins-mediated drug influx contribute to the intrinsic antimicrobial susceptibility and inhibit streptomycin resistance acquisition by natural transformation in *Helicobacter pylori*)

Dear Prof. Yanmei Zhang:

Your manuscript has been accepted, and I am forwarding it to the ASM Journals Department for publication. You will be notified when your proofs are ready to be viewed.

Sincerely,

Beile Gao
Editor, Microbiology Spectrum

Journals Department
Supplemental Figures and Table: Accept
Supplemental Dataset: Accept
Supplemental Table S5: Accept
Supplemental Table S7: Accept
Supplemental Table S3: Accept
Supplemental Table S6: Accept
Supplemental Table S4: Accept
Supplemental Material: Accept